# Learning with Mixture of Prototypes for Out-of-Distribution Detection

**Haodong Lu[1], Dong Gong[1]*, Shuo Wang[2], Jason Xue[2], Lina Yao[1,2], Kristen Moore[2]**
[1]University of New South Wales, [2]CSIRO's Data61
{haodong.lu, dong.gong}@unsw.edu.au,
{shuo.wang, jason.xue, lina.yao, kristen.moore}@data61.csiro.au

## Abstract

Out-of-distribution (OOD) detection aims to detect testing samples far away from the in-distribution (ID) training data, which is crucial for the safe deployment of machine learning models in the real world. Distance-based OOD detection methods have emerged with enhanced deep representation learning. They identify unseen OOD samples by measuring their distances from ID class centroids or prototypes. However, existing approaches learn the representation relying on oversimplified data assumptions, *e.g.*, modeling ID data of each class with one centroid class prototype or using loss functions not designed for OOD detection, which overlook the natural diversities within the data. Naively enforcing data samples of each class to be compact around only one prototype leads to inadequate modeling of realistic data and limited performance. To tackle these issues, we propose **P**rototyp**A**l **L**earning with a **M**ixture of prototypes (PALM) which models each class with multiple prototypes to capture the sample diversities, and learns more faithful and compact samples embeddings to enhance OOD detection. Our method automatically identifies and dynamically updates prototypes, assigning each sample to a subset of prototypes via reciprocal neighbor soft assignment weights. To learn embeddings with multiple prototypes, PALM optimizes a maximum likelihood estimation (MLE) loss to encourage the sample embeddings to be compact around the associated prototypes, as well as a contrastive loss on all prototypes to enhance intra-class compactness and inter-class discrimination at the prototype level. Compared to previous methods with prototypes, the proposed mixture prototype modeling of PALM promotes the representations of each ID class to be more compact and separable from others and the unseen OOD samples, resulting in more reliable OOD detection. Moreover, the automatic estimation of prototypes enables our approach to be extended to the challenging OOD detection task with unlabelled ID data. Extensive experiments demonstrate the superiority of PALM over previous methods, achieving state-of-the-art average AUROC performance of 93.82 on the challenging CIFAR-100 benchmark. Code is available at https://github.com/jeff024/PALM.

## 1 Introduction

Deep learning (DL) plays a crucial role in many real-world applications, such as autonomous driving (Huang et al., 2020), medical diagnosis (Zimmerer et al., 2022), and cyber-security (Nguyen et al., 2022). When deployed in realistic open-world scenarios (Drummond & Shearer, 2006), deep neural networks (DNNs) trained on datasets that adhere to closed-world assumptions (He et al., 2015), commonly known as in-distribution (ID) data, tend to struggle when faced with testing samples that significantly deviate from the training distribution, referred to as out-of-distribution (OOD) data. A trustworthy learning system should be aware of OOD samples instead of naively assuming all input to be ID. In recent years, there has been significant research focus on the OOD detection task (Drummond & Shearer, 2006), aiming to accurately distinguish between OOD and ID inputs. This critical endeavor helps to ensure the secure and reliable deployment of DNN models.

Since OOD samples are unseen during training, the key challenge is to obtain a model and an associated OOD detection criterion, based on only ID samples. Various OOD detection methods

---

*D. Gong is the corresponding author. This work was partially supported by an ARC DECRA Fellowship (DE230101591) awarded to D. Gong. H. Lu is affiliated with CSIRO Data61 through a PhD scholarship.

have been developed recently, including confidence score based methods (Hendrycks & Gimpel, 2016; Lee et al., 2018; Liang et al., 2018; Liu et al., 2020; Wang et al., 2021; Sun et al., 2021; Huang et al., 2021; Wang et al., 2022), density-based methods (Kingma & Dhariwal, 2018; Du & Mordatch, 2019; Grathwohl et al., 2020; Ren et al., 2019; Xiao et al., 2020; Cai & Li, 2023), and distance-based methods (Tack et al., 2020; Tao et al., 2023; Sun et al., 2022; Du et al., 2022a; Ming et al., 2023; Lee et al., 2018; Sehwag et al., 2021). Promising distance-based methods leverage the capability of DNNs to extract feature embeddings and identify OOD samples by measuring the distances between the embeddings and the centroids or prototypes of ID classes (Lee et al., 2018; Sehwag et al., 2021; Ming et al., 2023; Tao et al., 2023). They are effective in many scenarios, compared with other methods that overconfidently misclassify OOD data as ID (Hendrycks & Gimpel, 2016; Liang et al., 2018), or suffer from difficulty in training generative models (Ren et al., 2019; Grathwohl et al., 2020).

Distance-based OOD detection methods aim to learn informative feature embeddings and utilize distance metrics, such as Mahalanobis (Lee et al., 2018; Sehwag et al., 2021) or KNN distance (Sun et al., 2022), during testing to identify OOD samples. Recent advances in distance-based methods (Tack et al., 2020; Sehwag et al., 2021) use off-the-shelf contrastive loss (Chen et al., 2020; Khosla et al., 2020) to shape the embedding space, which is designed for classification and does not take OOD data into consideration. On top of that, Ming et al. (2023); Du et al. (2022a) shape the hyperspherical embedding space (Wang & Isola, 2020; Wang & Liu, 2021; Du et al., 2022a; Ming et al., 2023) of in-distribution (ID) data with class-conditional von Mises-Fisher (vMF) distributions (Mardia et al., 2000), enforcing ID samples of the same class to be compactly embedded around the prototype of its class. However, in Sec. 4.2 we show that naively modeling all samples of each class with only one single prototype leads to restricted modeling capability, where diverse patterns within each class cannot be well represented, leading to the confusion of ID samples and the OOD samples seen in testing. This lack of comprehensive representation further diminishes the compactness surrounding each prototype, thereby resulting in diminished performance.

In this paper, we propose a novel distance-based OOD detection method, called **P**rototypic**A**l **L**earning with a **M**ixture of prototypes (PALM) to learn high-quality hyperspherical embeddings of the data. To capture the natural diversities within each class, we model the hyperspherical embedding space (Sehwag et al., 2021; Sun et al., 2022; Du et al., 2022a; Ming et al., 2023) of each class by a mixture vMF distributions with multiple prototypes, where each prototype represents a subset of thee most similar data samples. Instead of naively enforcing data samples of each class to be compact around a single prototype (Du et al., 2022a; Ming et al., 2023; Tao et al., 2023), we encourage a more compact cluster around each one of the multiple prototypes, inherently encourage better ID-OOD discrimination. Specifically, we automatically estimate the reciprocal *assignment weights* between data samples and prototypes, and dynamically update the prototypes guided by these weights. To ensure each sample a high probability of assignment to the corresponding vMF distribution conditioned on each prototype, we apply a *MLE loss* to minimize the distance between each sample embedding and the prototype, similar to (Ming et al., 2023; Li et al., 2021; Caron et al., 2020). Furthermore, we propose a *contrastive loss on the prototypes* to further enhance the intra-class compactness at the prototype/cluster level and simultaneously encourage inter-class discrimination. Our main contributions are summarized as follows:

- We propose a novel distance-based OOD detection method, *i.e.*, PALM, which regularizes the representation learning in hyperspherical embedding space. Unlike previous methods with oversimplified assumptions, we use more realistic modeling with a mixture of prototypes to formulate and shape the embedding space, leading to better ID-OOD discrimination.

- In PALM, we propose a prototypical learning framework to learn the mixture prototypes automatically. Samples are softly assigned to prototypes using specifically designed methods. PALM uses a MLE loss between samples and prototypes, as well as a contrastive loss on all prototypes to enhance intra-class compactness and inter-class discrimination.

- Extensive experiments and in-depth analyses show the effectiveness of PALM on OOD detection. In addition to the standard labelled setting, the automatic prototype learning enables PALM to be easily extended to unsupervised OOD detection with promising results.

## 2 RELATED WORK

**Out-of-distribution detection.** The problem of OOD detection was first posed by Nguyen et al. (2015), to address the limitation that DNN models were found to tend to generate overconfident results. This problem has drawn increasing research interest ever since. To overcome this issue,

various OOD detection methods have been proposed like score-based methods (Hendrycks & Gimpel, 2016; Lee et al., 2018; Liang et al., 2018; Liu et al., 2020; Wang et al., 2021; Sun et al., 2021; Huang et al., 2021; Wang et al., 2022), generative-based methods (Ryu et al., 2018; Kong & Ramanan, 2021) and distance-based methods (Tack et al., 2020; Tao et al., 2023; Sun et al., 2022; Du et al., 2022a; Ming et al., 2023; Lee et al., 2018; Sehwag et al., 2021). Score-based methods derive scoring functions for model outputs in output space including confidence-based score (Hendrycks & Gimpel, 2016; Liang et al., 2018; Sun et al., 2021; Wang et al., 2022; Wei et al., 2022), classification score from a learned discriminator (Kong & Ramanan, 2021), energy-based score (Liu et al., 2020; Wang et al., 2021) and gradient-based score (Huang et al., 2021). Building upon the idea that OOD samples should be far away from known ID data centroids or prototypes in the embedding space, recent distance-based methods have shown promising results by computing distances to the nearest training sample (Tack et al., 2020) or nearest prototype (Tao et al., 2023), KNN distance (Sun et al., 2022; Du et al., 2022a; Ming et al., 2023) and Mahalanobis distance (Lee et al., 2018; Sehwag et al., 2021). In addition to that, only a few methods (Tack et al., 2020; Sun et al., 2022) explore the potential of developing OOD detection models using completely unlabeled training data. In this work, we extend our framework to be able to work with unlabeled training data.

**Representation learning for OOD detection.** Building on the recent success of contrastive representation learning methods such as SimCLR (Chen et al., 2020) and SupCon (Khosla et al., 2020), recent distance-based methods (Tack et al., 2020; Sehwag et al., 2021; Sun et al., 2022) have demonstrated the successful application of these methods to OOD detection, despite their training objectives not being specifically designed for this task. Notably, features obtained from SupCon (Khosla et al., 2020) have been used to compute distance metrics, such as the Mahalanobis distance (Sehwag et al., 2021) and KNN distance (Sun et al., 2022), for OOD detection. These approaches outperform previous distance-based methods that derive features trained from standard cross entropy (Lee et al., 2018). Methods like VOS (Du et al., 2022b) and NPOS (Tao et al., 2023) have taken a different approach by synthesizing OOD data samples to regularize the model's decision boundary between ID and OOD data. Recent works (Du et al., 2022a; Ming et al., 2023) that model the data as vMF distribution (Mardia et al., 2000) provide simple and clear interpretation of the hyperspherical embeddings. Specifically, Ming et al. (2023) propose a regularization strategy to ensure that all samples are compactly located around their class's corresponding single prototype.

**Contrastive learning and prototypical learning.** Contrastive representation learning methods treat each sample as an individual class, and bring together multiple views of the same input sample while pushing away from other samples. This effectively enhances the discriminative properties of the learned representations, enabling these approaches to take the lead in learning powerful feature representations in unsupervised (Wu et al., 2018; Oord et al., 2018; He et al., 2020; Chen et al., 2020; Robinson et al., 2021), semi-supervised (Assran et al., 2021) and supervised settings (Khosla et al., 2020). The fundamental properties and effectiveness of contrastive loss in hyperspherical space have been investigated in studies (Wang & Isola, 2020; Wang & Liu, 2021). Other methods learn feature representations by modeling the relations of samples to cluster centroids (Caron et al., 2018) or prototypes (Snell et al., 2017). On top of contrastive learning, Li et al. (2021) integrate prototypical learning that additionally contrasts between samples and prototypes obtained through offline clustering algorithms. The introduced prototypes benefit the representation ability but require all training samples for cluster assignments, leading to training instability due to label permutations (Xie et al., 2022). Unlike the existing prototypical learning methods focusing on basic classification tasks with generic designs, the proposed method uses a novel and specifically designed mixture of prototypes model for OOD detection.

## 3 METHOD

Let $\mathcal{X}$ and $\mathcal{Y}^{\text{id}}$ denote the input and label space of the ID training data given to a machine learning model, respectively. For example, $\mathcal{X} := \mathbb{R}^n$ and $\mathcal{Y}^{\text{id}} := \{1, ..., C\}$ are the input image space and the label space for multi-label image classification. A machine learning method can access the ID training dataset $\mathcal{D}^{\text{id}} = \{(\mathbf{x}_i, y_i)\}$ drawn i.i.d. from the joint distribution $\mathcal{P}_{\mathcal{X} \times \mathcal{Y}^{\text{id}}}$, and assumes the same distribution (*e.g.*, same label space) in training and testing under a closed-world setting.

The aim of OOD detection is to identify whether a testing sample $\mathbf{x} \in \mathcal{X}$ is from ID or not (*i.e.*, OOD). In a typical scenario of classification (Hendrycks & Gimpel, 2016), the OOD data are from unknown classes $y \notin \mathcal{Y}^{\text{id}}$, *i.e.*, $\mathcal{Y}^{\text{id}} \cap \mathcal{Y}^{\text{ood}} = \emptyset$. By letting $\mathcal{P}_{\mathcal{X}}^{\text{id}}$ denote the marginal distribution of $\mathcal{P}_{\mathcal{X} \times \mathcal{Y}^{\text{id}}}$ on $\mathcal{X}$, an input $\mathbf{x}$ is identified as OOD according to $\mathcal{P}^{\text{id}}(\mathbf{x}) < \sigma$, where threshold $\sigma$ is a level

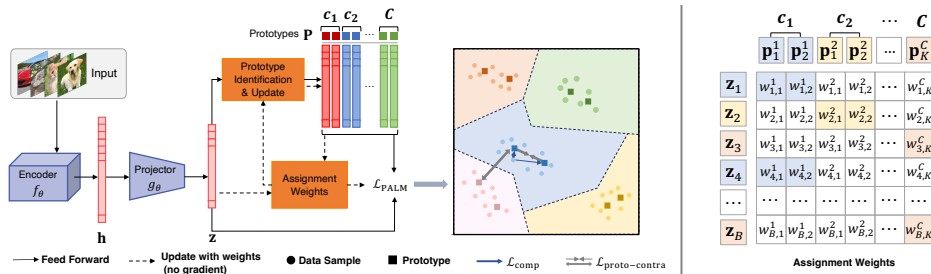

Figure 1: Overview of our proposed framework of prototypical learning with a mixture of prototypes (**PALM**). We regularize the embedding representation space via the proposed mixture of prototype modeling. We propose (1) optimizing an MLE loss to encourage the sample embeddings to be compact around their associated prototypes, and (2) minimizing *prototype contrastive loss* to regularize the model at the prototype level. We visualize the calculation of assignment weights on the right.

set parameter determined by the false ID detection rate (*e.g.*, 0.05) (Ming et al., 2023; Chen et al., 2017). Note that OOD samples are not seen/available during training, and the ID training data are labelled. In the unsupervised/unlabelled OOD/outlier detection (Sehwag et al., 2021; Yang et al., 2021) setting, the model can only access unlabelled ID data $\mathcal{D}^{\text{id}} = \{\mathbf{x}_i\}$.

### 3.1 OVERVIEW OF THE PROPOSED METHOD

The proposed method PALM consists of three main components, as shown in Fig. 1. A DNN *encoder* $f_\theta : \mathbf{X} \to \mathbb{R}^E$ is used to extract feature embeddings $\mathbf{h} \in \mathbb{R}^E$ from the input $\mathbf{x}$ with $\mathbf{h} = f_\theta(\mathbf{x})$. To regularize the representation learning, a *projector* $g_\phi : \mathbb{R}^E \to \mathbb{R}^D$ followed by normalization is used to project the high-dimensional embedding $\mathbf{h}$ to a lower-dimensional hyperspherical embedding $\mathbf{z}$ via $\mathbf{z}' = g_\phi(\mathbf{h})$ and $\mathbf{z} = \mathbf{z}'/\|\mathbf{z}'\|_2$. Relying on the mixture modeling of the hyperspherical embeddings, the proposed *prototypical learning module* with the associated losses is used to shape the embedding space and regularize the learning of $f_\theta$ and $g_\phi$, which will produce discriminative embeddings for OOD detection. Distance metrics (Sehwag et al., 2021; Sun et al., 2022) will be applied to embeddings $\mathbf{h}$ to determine the OOD detection scores, as in other distance/representation-based methods (Tack et al., 2020; Tao et al., 2023; Du et al., 2022a).

### 3.2 MODELING HYPERSPHERICAL EMBEDDINGS WITH MIXTURE MODELS

We formulate the embedding space with hyperspherical model, considering the its benefits in representation learning mentioned above (Wang & Isola, 2020; Khosla et al., 2020). The projected embeddings $\mathbf{z}$ lying on the unit sphere ($\|\mathbf{z}\|^2 = 1$) can be naturally modeled using the von Mises-Fisher (vMF) distribution (Mardia et al., 2000; Wang & Isola, 2020). Generally, the whole embedding space can be modeled as a mixture of vMF distributions, where each is defined by a mean $\mathbf{p}_k$ and a concentration parameter $\kappa$: $p_D(\mathbf{z}; \mathbf{p}_k, \kappa) = Z_D(\kappa) \exp\left(\kappa \mathbf{p}_k^\top \mathbf{z}\right)$, where $\mathbf{p}_k \in \mathbb{R}^D$ is the the $k$-th prototype with unit norm, $\kappa \geq 0$ represents the tightness around the mean, and $Z_D(\kappa)$ is the normalization factor.

Prior works use single class-conditional vMF distribution with one prototype to represent the samples within a specific class (Du et al., 2022a; Ming et al., 2023). In the whole embedding space, each sample is assigned to and pushed to the single prototype of its class. However, uniformly enforcing all samples to be close to one prototype may be unrealistic for complex data, leading to insufficient representative capability, non-compact embeddings and confusions between ID and OOD, as visualized in Fig. 5. We thus propose to model each class with a mixture of vMF distributions by multiple prototypes. For each class $c$, we define $K$ prototypes $\mathbf{P}^c = \{\mathbf{p}_k^c\}_{k=1}^K$ corresponding to a mixture of vMF. Each $\mathbf{z}_i$ is assigned to the prototypes via assignment weights $\mathbf{w}_i^c \in \mathbb{R}^K$. The probability density for a sample $\mathbf{z}_i$ in class $c$ is defined as a mixture model:

$$p(\mathbf{z}_i; \mathbf{w}_i^c, \mathbf{P}^c, \kappa) = \sum_{k=1}^K w_{i,k}^c Z_D(\kappa) \exp(\kappa \mathbf{p}_k^{c\top} \mathbf{z}_i), \tag{1}$$

where $w_{i,k}^c$ denotes the $k$-th element of $\mathbf{w}_i^c$. We define the same number of prototypes, *i.e.*, same $K$, for each class. In PALM, the assignment weights are decided according to the adjacency relationship between the samples and the estimated prototypes, as discussed in the following. Given the probability model in Eq. (1), an embedding $\mathbf{z}_i$ is assigned to a class $c$ with the following normalized probability:

$$p(y_i = c | \mathbf{z}_i; \{\mathbf{w}_i^j, \mathbf{P}^j, \kappa\}_{j=1}^C) = \frac{\sum_{k=1}^K w_{i,k}^c \exp(\mathbf{p}_k^{c\top} \mathbf{z}_i / \tau)}{\sum_{j=1}^C \sum_{k'=1}^K w_{i,k'}^j \exp(\mathbf{p}_{k'}^{j\top} \mathbf{z}_i / \tau)}, \tag{2}$$

where $\tau = 1/\kappa$ is analogous to the temperature parameter in our MLE loss.

### 3.3 PROTOTYPICAL LEARNING WITH MIXTURE OF PROTOTYPES

Relying on the class-conditional mixture vMF distributions of different classes, we can regularize the representation learning in the hyperspherical embedding space with the prototypes as anchors. Sharing a similar motivation to prior works, we optimize the derived end-to-end trainable objectives to 1) let each sample be assigned to the correct class with a higher probability compared to the incorrect classes, and 2) learn more informative representations for discriminating different classes, as well as ID and OOD samples. In PALM, we learn the (randomly initialized) prototypes and the assignment weights dynamically, and conduct the prototypical learning via optimizing the objectives relying on the mixture prototypes derived from the vMF mixture distributions.

**Training objectives.** Given the training data, we can perform maximum likelihood estimation (MLE) according to Eq. (1) by solving the problem $\max_{\theta,\phi} \prod_{i=1}^{N} p(y_i = c | \mathbf{z}_i, \{\mathbf{w}_i^c, \{\mathbf{p}_k^c, \kappa\}_{k=1}^{K}\}_{j=1}^{C})$, where $\mathbf{z}_i$ is the hyperspherical embedding obtained with $f_\theta$ and $g_\phi$. By taking the negative log-likelihood, the optimization problem can be equivalently rewritten as:

$$\mathcal{L}_{\text{MLE}} = -\frac{1}{N} \sum_{i=1}^{N} \log \frac{\sum_{k=1}^{K} w_{i,k}^{y_i} \exp\left(\mathbf{p}_k^{y_i \top} \mathbf{z}_i / \tau\right)}{\sum_{c=1}^{C} \sum_{k'=1}^{K} w_{i,k'}^{c} \exp\left(\mathbf{p}_{k'}^{c \top} \mathbf{z}_i / \tau\right)}, \tag{3}$$

where $y^i$ represents the class index of the sample $\mathbf{x}_i$, and $\tau$ is the temperature parameter. This *MLE loss* encourages samples to be close to the proper prototypes belonging to their own classes. Instead of forcing all samples in one class to be compact around one prototype as in (Ming et al., 2023; Du et al., 2022a), PALM assigns each sample to specific prototypes according to the assignment weights. The mixture prototypes enable diverse samples to be better represented by specific prototypes.

The MLE loss in Eq. (3) only encourages the compactness between samples and the prototypes. To further shape the embedding space, we encourage intra-class compactness and inter-class discrimination at the prototype level. To do this, we propose the *prototype contrastive loss*, which relies on the class information of the prototypes as an implicit supervision signal (Khosla et al., 2020; Sehwag et al., 2021):

$$\mathcal{L}_{\text{proto-contra}} = -\frac{1}{CK} \sum_{c=1}^{C} \sum_{k=1}^{K} \log \frac{\sum_{k'=1}^{K} \mathbb{1}(k' \neq k) \exp(\mathbf{p}_k^{c \top} \mathbf{p}_{k'}^{c} / \tau)}{\sum_{c'=1}^{C} \sum_{k''=1}^{K} \mathbb{1}(k'' \neq k, c' \neq c) \exp(\mathbf{p}_k^{c \top} \mathbf{p}_{k''}^{c'} / \tau)}, \tag{4}$$

where $\mathbb{1}()$ is an indicator function for avoiding contrasting between the same prototype. As discussed in the following, the prototypes are updated with the exponential moving average (EMA) technique. Thus the loss in Eq. (4) is mainly used to regularize the sample embeddings through the connection between samples and the prototypes.

The overall training objective of PALM can be formally defined as:

$$\mathcal{L}_{\text{PALM}} = \mathcal{L}_{\text{MLE}} + \lambda \mathcal{L}_{\text{proto-contra}}, \tag{5}$$

where $\lambda > 0$ is the weight to control the balance between these two loss functions.

**Soft prototype assignment with reciprocal neighbor relationships.** The assignment weights are mainly used for calculating the MLE loss in Eq. (3) and updating the prototypes as discussed in the following, which are both in the training process with label information available. In the *class-conditional* mixture modeling, for each sample, we consider the assignment within its classes. Given a sample and the mixture prototypes of its class, we assign it to the prototypes relying on the adjacency information. Specifically, we first obtain the assignment weights by considering the global information with an online soft cluster assignment method in (Caron et al., 2020). Given $K$ prototypes $\mathbf{P}^c$ of class $c$ and a batch of $B$ embeddings $\mathbf{Z}$, having $B^c$ samples $\mathbf{Z}^c$ in class $c$, we can obtain the assignment weights packed in a matrix as:

$$\mathbf{W}^c = \text{diag}(\mathbf{u}) \exp\left(\frac{\mathbf{P}^{c \top} \mathbf{Z}^c}{\epsilon}\right) \text{diag}(\mathbf{v}), \tag{6}$$

where $\mathbf{u} \in \mathbb{R}_+^K$ and $\mathbf{v} \in \mathbb{R}_+^{B^c}$ are nonnegative renormalization vectors calculated for obtaining the solution (Cuturi, 2013), and diag() denotes the diagonal matrix of a vector. They can be obtained

through efficient matrix multiplications using the iterative Sinkhorn-Knopp algorithm (Cuturi, 2013), and we show the efficiency of this approximation in Appendix C.1 in detail.

**Assignment Pruning.** Eq. (6) assigns each sample to the prototypes with soft assignment weights. Some prototypes far away from a sample can also be assigned with small weights. Although every prototype can be a neighbor of a sample described by a weight (reflecting the distance), a prototype may not be a proper neighbor of a sample, considering it mainly describes another cluster of samples very close to it. We consider further refining the assignments obtained via Eq. (6), considering the potential mutual neighbor relationship from a local perspective. For simplicity, instead of producing complicated mutual neighbor analyses, we conduct pruning on the assignment weights of each sample via top-K shrinkage operation: $w_{i,k}^c := \mathbb{1}[w_{i,k}^c > \beta] * w_{i,k}^c$, where $\beta$ is the K-th largest assignment weight, and $\mathbb{1}[w_{i,k}^c > \beta] \in \{0, 1\}$ is an indicator function. We studied how the hyperparameter $K$ for the top-K selection will influence the performance in Sec. 4.3, as shown in Fig. 4(a).

**Prototype updating.** As mentioned previously, to ensure optimal performance during the optimization of network parameters, we implemented the widely-used exponential moving average (EMA) (Li et al., 2020) technique to update the prototype values. This enables model parameters ($f_\theta, h_\theta$) and prototypes $\mathbf{P}$ to be updated asynchronously. The EMA technique allows us to iteratively update the prototype values while maintaining a consistent assignment, which helps to smooth out training. This approach is aimed at avoiding sub-optimal solutions that may arise during training due to fluctuations in the prototype values. Given a batch of $B$ data samples, we formally denote the update rule as:

$$\mathbf{p}_k^c := \texttt{Normalize}(\alpha\mathbf{p}_k^c + (1 - \alpha)\sum_{i=1}^{B} \mathbb{1}(y_i = c)w_{i,k}^c\mathbf{z}_i), \tag{7}$$

where $\mathbb{1}()$ is an indicator function that ensures only selecting samples belonging to the same class for updating. Subsequently, we renormalize prototypes to the unit sphere for future optimization, ensuring that the distances between prototypes and embeddings remain meaningful and interpretable.

**Extension to unsupervised OOD detection.** In PALM, the prototypes and the assignments in the class-conditional mixture model can be automatically learned and estimated. Benefiting from this learning paradigm, we can extend PALM to the unsupervised OOD detection without any class labels available during training, with minor adjustments. Specifically, we do not use the label information and release the supervised learning version of prototype contrastive loss into an unsupervised learning version. Even with such simple modifications, PALM can perform well on unsupervised OOD setting, showing the potential. More details are left in Appendix A.4.

## 4 EXPERIMENTS

**Datasets and training details.** We use the standard `CIFAR-100` and `CIFAR-10` dataset (Krizhevsky et al., 2009) as our ID training dataset, and report OOD detection performance on a series of natural image datasets including `SVHN` (Netzer et al., 2011), `Places365` (Zhou et al., 2017), `LSUN` (Yu et al., 2015), `iSUN` (Xu et al., 2015), and `Textures` (Cimpoi et al., 2014). We further assess the performance of PALM on *Near-OOD detection* benchmarks (Tack et al., 2020; Sun et al., 2022), as outlined in Appendix C. In our main experiments, we use ResNet-34 as our backbone model for the CIFAR-100 ID training dataset and ResNet-18 for CIFAR-10, along with a two-layer MLP projector that projects to a 128-dimensional unit sphere following (Sehwag et al., 2021; Sun et al., 2022; Ming et al., 2023; Tao et al., 2023). We train the model using stochastic gradient descent with momentum 0.9 and weight decay $10^{-6}$ for 500 epochs. We employ the same initial learning rate of 0.5 with cosine learning rate scheduling (Loshchilov & Hutter, 2017; Misra & Maaten, 2020). By default, we maintain 6 prototypes for each class and select 5 closest prototypes during the pruning phase. To ensure consistent assignment between iterations, we use a large momentum $\alpha$ of 0.999 as the default value for prototype update. For *unsupervised OOD detection*, none of the class labels for ID training samples are available, presenting a considerably more challenging benchmark. More experimental details are provided in Appendix B and more experiment results Appendix C.

**OOD detection scoring function.** Given that our approach is designed to learn compact representations, we select the widely-used distance-based OOD detection method of Mahalanobis score (Lee et al., 2018; Sehwag et al., 2021). In line with standard procedure (Sehwag et al., 2021; Sun et al., 2022; Ming et al., 2023), we leverage the feature embeddings from the penultimate layer for distance metric computation. For completeness, we also compare our method using the recently proposed distance metric of KNN (Sun et al., 2022).

Table 1: OOD detection performance on methods trained on **labeled** CIFAR-100 as ID dataset using backbone network of ResNet-34. ↓ means smaller values are better and ↑ means larger values are better. **Bold** numbers indicate superior results.

| Methods | SVHN | | Places365 | | OOD Datasets LSUN | | iSUN | | Textures | | Average | |
|---|---|---|---|---|---|---|---|---|---|---|---|---|
| | FPR↓ | AUROC↑ | FPR↓ | AUROC↑ | FPR↓ | AUROC↑ | FPR↓ | AUROC↑ | FPR↓ | AUROC↑ | FPR↓ | AUROC↑ |
| MSP | 78.89 | 79.80 | 84.38 | 74.21 | 83.47 | 75.28 | 84.61 | 74.51 | 86.51 | 72.53 | 83.57 | 75.27 |
| Vim | 73.42 | 84.62 | 85.34 | 69.34 | 86.96 | 69.74 | 85.35 | 73.16 | 74.56 | 76.23 | 81.13 | 74.62 |
| ODIN | 70.16 | 84.88 | 82.16 | 75.19 | 76.36 | 80.1 | 79.54 | 79.16 | 85.28 | 75.23 | 78.70 | 78.91 |
| Energy | 66.91 | 85.25 | 81.41 | 76.37 | 59.77 | 86.69 | 66.52 | 84.49 | 79.01 | 79.96 | 70.72 | 82.55 |
| VOS | 43.24 | 82.8 | 76.85 | 78.63 | 73.61 | 84.69 | 69.65 | 86.32 | 57.57 | 87.31 | 64.18 | 83.95 |
| CSI | 44.53 | 92.65 | 79.08 | 76.27 | 75.58 | 83.78 | 76.62 | 84.98 | 61.61 | 86.47 | 67.48 | 84.83 |
| SSD+ | 31.19 | 94.19 | 77.74 | 79.90 | 79.39 | 85.18 | 80.85 | 84.08 | 66.63 | 86.18 | 67.16 | 85.91 |
| kNN+ | 39.23 | 92.78 | 80.74 | 77.58 | 48.99 | 89.30 | 74.99 | 82.69 | 57.15 | 88.35 | 60.22 | 86.14 |
| NPOS | 10.62 | 97.49 | 67.96 | 78.81 | 20.61 | 92.61 | 35.94 | 88.94 | **24.92** | 91.35 | 32.01 | 89.84 |
| CIDER | 12.55 | 97.83 | 79.93 | 74.87 | 30.24 | 92.79 | 45.97 | 88.94 | 35.55 | 92.26 | 40.85 | 89.34 |
| **PALM (ours)** | **3.29** | **99.23** | **64.66** | **84.72** | **9.86** | **98.01** | **28.71** | **94.64** | 33.56 | **92.49** | **28.02** | **93.82** |

Table 2: *Unsupervised OOD detection* performance on methods trained on **unlabeled** CIFAR-100 as ID dataset using backbone network of ResNet-34.

| Methods | SVHN | | Places365 | | OOD Datasets LSUN | | iSUN | | Textures | | Average | |
|---|---|---|---|---|---|---|---|---|---|---|---|---|
| | FPR↓ | AUROC↑ | FPR↓ | AUROC↑ | FPR↓ | AUROC↑ | FPR↓ | AUROC↑ | FPR↓ | AUROC↑ | FPR↓ | AUROC↑ |
| SimCLR+KNN | 61.21 | 84.92 | 81.46 | 72.97 | 69.65 | 77.77 | 83.35 | 70.39 | 78.49 | 76.75 | 74.83 | 76.56 |
| SSD | 60.13 | 86.40 | 79.05 | 73.68 | 61.94 | 84.47 | 84.37 | 75.58 | 71.91 | 83.35 | 71.48 | 80.70 |
| SimCLR | 52.63 | 91.52 | **77.51** | **76.01** | 31.28 | 94.05 | 90.90 | 66.51 | 70.07 | 81.81 | 64.48 | 81.98 |
| CSI | 14.47 | 97.14 | 86.23 | 66.93 | 34.12 | 94.21 | 87.79 | 80.15 | 45.16 | **92.13** | 53.55 | 86.11 |
| **PALM (ours)** | **13.86** | **97.53** | 85.63 | 69.46 | **21.28** | **95.95** | **53.43** | **89.06** | **42.62** | 88.33 | **43.37** | **88.07** |

**Evaluation metrics.** To demonstrate the effectiveness of PALM, we report three commonly used evaluation metrics: (1) the false positive rate (FPR) of OOD samples when the true positive rate of ID samples is at 95%, (2) the area under the receiver operating characteristic curve (AUROC), and (3) ID classification accuracy (ID ACC).

## 4.1 MAIN RESULTS

**PALM outperforms previous supervised methods by a large margin.** Table 1 shows the experimental results based on the standard setting of using CIFAR-100 as ID data and other datasets as unseen OOD data. For a fair comparison, all results are obtained using ResNet-34 trained on the ID CIFAR-100 dataset without access to any auxiliary outlier/OOD datasets. We compare our method with recent competitive methods, including MSP (Hendrycks & Gimpel, 2017), Vim (Wang et al., 2022), ODIN (Liang et al., 2018), Energy (Liu et al., 2020), VOS (Du et al., 2022b), CSI (Tack et al., 2020), SSD+ (Sehwag et al., 2021), kNN+ (Sun et al., 2022), NPOS (Tao et al., 2023), and CIDER (Ming et al., 2023).

Compared to previous distance-based methods such as SSD+ (Sehwag et al., 2021) and KNN+ (Sun et al., 2022), which employ contrastive loss designed for classification tasks, PALM outperforms them based on the regularization designed for OOD detection. All method performs not well for the Place365 OOD dataset due to its input being confusing with the ID data. NPOS achieves the best FPR on Textures as the OOD data, since it directly generates OOD samples to boost training. By modeling the embedding space with a mixture of prototypes, PALM achieves a notable **12.83%** reduction in average FPR compared to the most related work CIDER, which also models dependencies between input samples and prototypes. PALM outperforms the most recent work of NPOS, without the need to generate artificial OOD samples. Moreover, PALM achieves a new state-of-the-art level of AUROC performance. We also report the results on OpenOOD benchmark (Yang et al., 2022; Zhang et al., 2023) in Table 6 in Appendix C.

**PALM outperforms competitive unsupervised approaches.** As demonstrated in Table 2, our method in the unlabeled setting outperforms previous approaches, including SimCLR (Chen et al., 2020) + KNN (Sun et al., 2022), SimCLR (Chen et al., 2020; Tack et al., 2020), CSI (Tack et al., 2020), and SSD (Sehwag et al., 2021). Although our primary contribution does not specifically target this unlabeled setting, we still observe a significant performance boost, surpassing CSI on most of the datasets. It is worth noting that CSI utilizes an ensemble of different transformations and incurs 4 times the computational cost during both training and testing. Remarkably, our unsupervised method achieves on par or superior performance compared to our supervised prior works NPOS and

CIDER on various test datasets under the same training budget without any label information. More experiment results and discussions in Appendix C.2.

## 4.2 DISCUSSIONS

**PALM learns a more compact cluster around each prototype.** Comparing to previous distance-based methods that encourage samples of each class to be close to each other (Tack et al., 2020; Sehwag et al., 2021; Sun et al., 2022) or its single prototype (Ming et al., 2023), PALM considers a more realistic embedding space where samples are enforced to be close to its most similar prototype of its class and learn a more compact embedding space. In Fig.2 (top), we evaluate the embedding quality of PALM by calculating the cosine similarity of ID samples to their nearest prototype. We observe a more compact distribution with a higher number of similar samples and significantly fewer dissimilar samples. Notably, CIDER struggles with the representative ability of prototypes, where some ID samples even exhibit near-zero similarity. We also quantify the number of far ID samples, whose similarity is below 0.8, in Fig. 2 (bottom). To provide a comprehensive analysis, we additionally assess the compactness measurements of (Ming et al., 2023) in Fig. 2 (bottom), where it measures the average cosine similarity of input samples to its closest prototype.

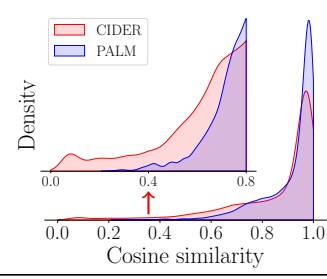

| Methods | Compactness ↓ (in degree) | Number of far ID samples (%) ↓ |
|---|---|---|
| CIDER | 31.08 | 25.79 |
| **PALM** | **24.21** | **15.71** |

Figure 2: Analysis of embedding quality for CIFAR-100 (ID) using CIDER and PALM. We examine the distance distribution (**top**), and evaluate compactness and the proportion of far ID samples (**bottom**).

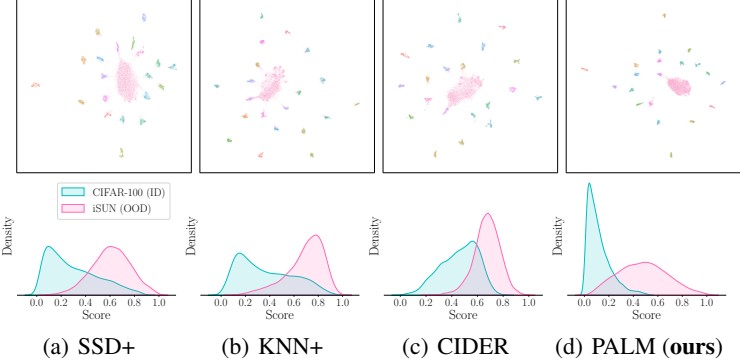

| (a) SSD+ | (b) KNN+ | (c) CIDER | (d) PALM (**ours**) |

Figure 3: UMAP (McInnes et al., 2018) visualization of the first 20 subclasses of ID (CIFAR-100) and all OOD (iSUN) samples plotted to the same embedding space for methods including (a) SSD+ (Sehwag et al., 2021) (b) KNN+ (Sun et al., 2022) (c) CIDER (Ming et al., 2023) and (d) PALM. The scores are obtained by scaling the distance metrics used by each method to $[0, 1]$ for visulization. We measure the area of overlapping sections between ID and OOD scores, as shown in Fig. 5.

**PALM enhances the separability between ID and OOD.** We visualize the embedding distribution of ID (CIFAR-100) and OOD (iSUN) samples (central cluster in pink) using UMAP (McInnes et al., 2018) in Fig. 3. In contrast to previous works that suffer from correlations between OOD samples and numerous ID clusters (Figs. 3(a)3(b)3(c)), PALM exhibits only a weak correlation with a single ID cluster (Fig. 3(d)). Notably, PALM further enhances the separability between ID and OOD samples by promoting a more noticeable distance between ID clusters less correlated to OOD samples.

Additionally, we estimate the distance density distribution of ID and OOD samples for each UMAP visualization in Fig. 3 and quantitatively measure the area of overlapping sections in Fig. 5. Our method significantly promotes a large separation between ID and OOD samples, suggesting a more effective OOD detection performance.

## 4.3 ABLATION STUDIES

**PALM generalizes well to various distance-based OOD detection scoring functions.** To demonstrate the effectiveness of our proposed representation learning framework for OOD detection, we perform ablation studies on the choice of distance metrics in Table 3. We tested

Table 3: Ablation on distance metrics selection for OOD detection. We evaluate two widely used distance metrics of KNN distance (K=300) and Mahalanobis distance.

| Distance Metrics | Methods | OOD Datasets | | | | | | | | | | Average | |
| | | SVHN | | Places365 | | LSUN | | iSUN | | Textures | | | |
| | | FPR↓ | AUROC↑ | FPR↓ | AUROC↑ | FPR↓ | AUROC↑ | FPR↓ | AUROC↑ | FPR↓ | AUROC↑ | FPR↓ | AUROC↑ |
| KNN | KNN+ | 39.23 | 92.78 | 80.74 | 77.58 | 48.99 | 89.30 | 74.99 | 82.69 | 57.15 | 88.35 | 60.22 | 86.14 |
| | NPOS | 10.62 | 97.49 | 67.96 | 78.81 | 20.61 | 92.61 | 35.94 | 88.94 | 24.92 | 91.35 | 32.01 | 89.84 |
| | CIDER | 12.55 | 97.83 | 79.93 | 74.87 | 30.24 | 92.79 | 45.97 | 88.94 | 35.55 | 92.26 | 40.85 | 89.34 |
| | **PALM (ours)** | **6.90** | **98.54** | **61.87** | **82.32** | **14.60** | **96.67** | **30.76** | **92.06** | 37.04 | 90.84 | **30.23** | **92.09** |
| Mahalanobis | SSD+ | 31.19 | 94.19 | 77.74 | 79.90 | 79.39 | 85.18 | 80.85 | 84.08 | 66.63 | 86.18 | 67.16 | 85.91 |
| | **PALM (ours)** | **3.29** | **99.23** | **64.66** | **84.72** | **9.86** | **98.01** | **28.71** | **94.64** | **33.56** | **92.49** | **28.02** | **93.82** |

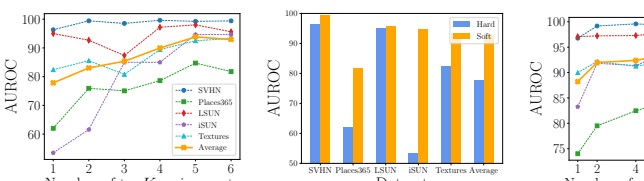

(a) Top-K assignment #.    (b) Assignment rules.    (c) Number of prototypes.    (d) Prototype updating.

Figure 4: Ablation studies on (a) pruning selection, (b) soft vs. hard assignments, (c) number of prototypes for each class and (d) prototype update procedure.

our method on two commonly used distance metrics, KNN distance (Sun et al., 2022; Ming et al., 2023; Tao et al., 2023) and Mahalanobis distance (Lee et al., 2018; Sehwag et al., 2021). Our method achieves superior performance on these two widely-used distance metrics.

**PALM improves large-scale OOD detection.** We demonstrate the performance of PALM on a more challenged large-scale benchmark in Fig. 6. We consider `ImageNet-100` (Tian et al., 2020) as our ID training dataset. Following (Huang & Li, 2021; Ming et al., 2023), we evaluate our OOD detection performance on test datasets `SUN` (Xiao et al., 2010), `Places` (Zhou et al., 2017), `Textures` (Cimpoi et al., 2014) and `iNaturalist` (Van Horn et al., 2018). More experiment details are left in Appendix B and experiment results in Appendix C.1.

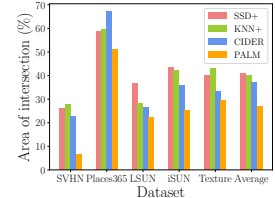

Figure 5: Area of overlapping sections between ID and OOD distance densities in percentage. Smaller numbers indicate superior results.

**Pruning less relevant assignments benefits the learning process.** Fig. 4(a) demonstrates the effectiveness of our introduced pruning procedure by top-K selection. We observe a significant improvement in performance when selecting more assignments, up to 5 out of the 6 available prototypes.

**Soft assignment helps model find optimal solutions.** In Fig. 4(b), we compare the performance between soft assignments and hard assignments. Soft assignments help preventing stuck in sub-optimal local minima.

**Ablation on number of prototypes.** We demonstrate the effects of varying the number of prototypes in Fig. 4(c) and observe an increase in performance as the number of prototypes increases. We find that defining only one prototype for each class, results in worse performance.

**EMA provides a more consistent cluster assignment.** As widely discussed in the literature (He et al., 2020; Grill et al., 2020; Ming et al., 2023), an asynchronous update between the model parameters of the two views or between model parameters and prototypes is fundamental to the success of recent representation learning methods. Here we also examine the effectiveness of the EMA update used for our prototypes in Fig. 4(d).

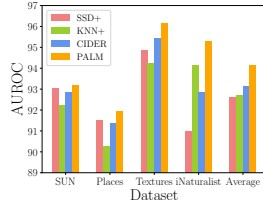

Figure 6: OOD detection performance on methods by fine-tuning pretrained ResNet-50 models on ImageNet-100.

## 5 CONCLUSION

In this work, we propose PALM, a novel prototypical learning framework with a mixture of prototypes that learns hyperspherical embeddings for OOD detection. By considering the complex underlying structure of data distributions, PALM model the embedding space by a mixture of multiple prototypes conditioned on each class, encouraging a more compact data distribution and demonstrating superior OOD detection performance. Moreover, we impose two extensions where we scale our method to large-scale datasets and unsupervised OOD detection. The limitation is that PALM needs to be manually assign the number of prototypes as a hyperparameter, which is left as future work.

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

APPENDIX

## A   MORE DETAILS OF THE PROPOSED METHOD

### A.1   ALGORITHM DETAILS

We present the training scheme of the proposed method, Prototypical Learning with Mixture of prototypes (PALM), in Algorithm 1. The algorithm maintains and updates prototypes, conducts the soft prototype assignment for the samples, optimizing the *MLE loss* and the *prototype contrastive loss*. Algorithm 1 summarizes the main operations in the training process and omits some calculation steps, *e.g.*, data augmentation.

In step 8 of Algorithm 1, the prototype assignment weights are calculated by considering the global information on the relationship between sample embeddings and the prototypes. The weights are then further refined, considering the mutual neighborhood relationship in step 9. By pruning the weights to keep top-K neighbor prototypes for each sample, the operation makes sure the samples and the assigned prototypes are not too far away from each other. Intuitively, the samples are "attached" to the prototypes in the EMA updating process. The prototypes work as "anchors" to back-propagate the gradients from optimizing the objectives to update the NN parameters, where the mixture of prototypes and the hyperspherical modeling regularize the NN representation learning. At the end of each iteration, the prototypes are detached from gradient calculation for the next round of updates.

---

**Algorithm 1:** Prototypical Learning with Mixture of prototypes (PALM)

**Input:** Training dataset $\mathcal{D}^{\text{id}}$ with $C$ classes, neural network encoder $f_\theta$, projection head $g_\phi$, loss weight, other hyperparameters, *e.g.*, number of prototypes $K$ for each class.

1  Initialize the NN parameters;
2  Initialize the class-conditional prototypes $\{\{\mathbf{p}_k^c\}_{k=1}^K\}_{c=1}^C$ for each class $c$;
    // No grad.  on initialized prototypes
3  **while** *training stopping conditions are not achieved* **do**
       // mini-batch iterations in each epoch:
4     **for** *iter = 1, 2, ...,* **do**
5        Sample a mini-batch of $B$ input samples $\{(\mathbf{x}_i, y_i)\}_{i=1}^B$;
6        Obtain projected embeddings for each input: $\mathbf{z}_i' = g_\phi(f_\theta(\mathbf{x}_i)), i \in \{1, ..., B\}$;
7        Normalize the embeddings to the unit sphere via $\mathbf{z}_i = \mathbf{z}_i'/\|\mathbf{z}_i'\|_2, i \in \{1, ..., B\}$;
        // Soft prototype assignment in Step 8-9:
8        Compute soft assignment weights for each set of the class-conditional mixture of prototypes $\{\mathbf{p}_k^c\}_{k=1}^K, c = 1, ..., C$, relying on Eq. (6) ;
9        Obtain the assignment weights via assignment weights pruning via Sec. 3.3;
10      Prototype updating via EMA in Eq. (7) (based on assignment weights);
        // Optimizing training objectives:
        // Grad.  back-propagating to NN parameters through the
          prototypes.
11      Calculate the objectives with $\mathcal{L}_{\text{assign}}$ and $\mathcal{L}_{\text{proto-contra}}$ in Eq. (3) and Eq. (4), respectively, and optimizing for updating network parameters;
12      Detach the updated prototypes from gradient calculation (for next round of updating);

---

### A.2   MORE DETAILS ON PROTOTYPE ASSIGNMENT AND UPDATING

During training, we assign each sample to the prototypes with soft assignment weights. In Sec. 3.3 of the main paper, we introduce how to calculate the assignment weights considering the global information and then refine the weights via pruning operation considering the local mutual neighborhood relationship. We put more details about the assignment weights calculation and prototype updating in the following.

**Estimation of the soft assignment weights.** Recall that we have $K$ prototypes $\mathbf{P}^c = \{\mathbf{p}_k^c\}_{k=1}^K$ for each class $c$, where $\mathbf{p}_k^c \in \mathbb{R}^D$. In mini-batch training, we obtain a batch of $B$ samples and the

corresponding projected embeddings $\{\mathbf{z}_i\}_{i=1}^B$, with $\mathbf{z} \in \mathbb{R}^D$, as shown in Algorithm 1. As introduced in Sec. 3.2, each sample/embedding is assigned to prototypes in its corresponding class with weights $\mathbf{w}^c \in \mathbb{R}^K$. Instead of assigning each sample to one prototype with a hard assignment, we use soft assignment weights to approach more smooth modeling. Experiments in Sec. 4.3 show the benefits of soft assignments as in Fig. 4(b). In each iteration of the mini-batch training, the prototype assignment aims to obtain the weights $\mathbf{w}^c$ for each sample.

To avoid the trivial solution where all samples may be assigned to the sample prototype (Caron et al., 2020; Gong et al., 2019; Zhou et al., 2022) in the online updating process, we encourage the samples in a batch (belonging to the same class) to be assigned to diverse prototypes (for the corresponding class), as the previous methods. As briefly introduced in the main paper, we define $\mathbf{Z}^c \in \mathbb{R}^{D \times B^c}$ to represent the concatenated embeddings of the class $c$, *i.e.*, $\mathbf{Z}^c = [\mathbf{z}_1, ..., \mathbf{z}_i, ..., \mathbf{z}_{B^c}]$, $\mathbf{P}^c \in \mathbb{R}^{D \times K}$ with $\mathbf{P}^c = [\mathbf{p}_1^c, ..., \mathbf{p}_i^c, ..., \mathbf{p}_K^c]$, and $\mathbf{W}^c = \mathbb{R}^{K \times B^c}$ for $\mathbf{W}^c = [\mathbf{w}_1^c, ..., \mathbf{w}_i^c, ..., \mathbf{w}_{B^c}^c]$, where $B^c$ denotes the number of samples belonging in to class $c$ in the batch. To encourage the diverse prototype assignment, we optimize $\mathbf{W}^c$ (for each class $c$) by solving the following problem to maximize the similarity of the assigned prototypes and the samples (Caron et al., 2020):

$$\max_{\mathbf{W}^c \in \mathcal{W}} \mathrm{Tr}(\mathbf{W}^{c\top}\mathbf{P}^{c\top}\mathbf{Z}^c) + \epsilon R(\mathbf{W}^c), \tag{8}$$

where $R(\cdot)$ is the entropy function as the regularize on $\mathbf{W}^c$, *i.e.*, $R(\mathbf{W}^c) = -\sum_{i,j} w_{i,j}^c \log w_{i,j}^c$ with $w_{i,j}^c$ to denote the $i, j$-th element in the matrix $\mathbf{W}^c$, $\epsilon$ is the regularization weight, $\mathcal{W}$ is the defined domain of $\mathbf{W}^c$. Following (Caron et al., 2020), $\mathcal{W}$ is defined as a *transportation polytope* (Asano et al., 2020), in the format of

$$\mathcal{W} = \{\mathbf{W} \in \mathbb{R}_+^{K \times B} | \mathbf{W}\mathbf{1}_B = \frac{1}{K}\mathbf{1}, \mathbf{W}^T\mathbf{1}_K = \frac{1}{B}\mathbf{1}_B\}, \tag{9}$$

where $\mathbf{1}_K$ and $\mathbf{1}_B$ denote the vector of ones in dimension of $K$ and $B$, respectively. It can be handled efficiently in Problem Eq. (8). Please find more details of the definition of the transportation polytope in (Caron et al., 2020; Asano et al., 2020; Cuturi, 2013). As introduced in the main paper, the soft assignment weights can be obtained by solving the problem in Eq. (8) as the following normalized exponential matrix (Cuturi, 2013):

$$\mathbf{W}^c = \mathtt{diag}(\mathbf{u}) \exp(\frac{\mathbf{P}^{c\top}\mathbf{Z}^c}{\epsilon}) \mathtt{diag}(\mathbf{v}), \tag{6}$$

where $\mathbf{u}$ and $\mathbf{v}$ are renormalization vectors in $\mathbb{R}^K$ and $\mathbb{R}^B$ obtained from a few steps of *Sinkhorn-Knopp* iterations (Cuturi, 2013; Caron et al., 2020). As shown in Lemma 2 in (Cuturi, 2013), the solution of $\mathbf{W}^c$ with the transportation polytope constraint in Eq. (9) is unique and can be obtained via the form of Eq. (6). $\mathbf{u}$ and $\mathbf{v}$ are two non-negative vectors for obtaining the solution, which can be computed with Shinkhorn's fixed point iteration $(\mathbf{u}, \mathbf{v}) \leftarrow (1/(K\mathbf{H}\mathbf{v}), 1/(B\mathbf{H}\mathbf{u}))$, where we use $\mathbf{H}$ to denote $\exp(\frac{\mathbf{P}^{c\top}\mathbf{Z}^c}{\epsilon})$ in Eq. (6). With further simplification, $\mathbf{u}$ and $\mathbf{v}$ can be obtained with a few iterations, as discussed in Algorithm 1 in (Cuturi, 2013). Please find the proof and more details in (Cuturi, 2013). $\mathtt{diag}(\mathbf{u})$ and $\mathtt{diag}(\mathbf{v})$ are the diagonal matrix of the two vectors $\mathbf{u}$ and $\mathbf{v}$. Through solving the problem in Eq. (8), the assignment weights are obtained by considering the similarity between embeddings and prototypes (as reflected by $\mathbf{P}^{c\top}\mathbf{Z}^c$) and the relationships between the prototypes and all the samples in a global viewpoint. In practice, the calculations for different classes can be conducted simultaneously and in parallel.

### A.3 MORE STEPS FOR OBTAINING EQ. (2)

The probability density for a $\mathbf{z}_i$ in class $c$ is defined as the mixture model in Eq. (1) in the main paper:

$$p(\mathbf{z}_i; \mathbf{w}_i^c, \{\mathbf{p}_k^c, \kappa\}_{k=1}^K) = \sum_{k=1}^K w_{i,k}^c Z_D(\kappa) \exp(\kappa \mathbf{p}_k^{c\top}\mathbf{z}_i),$$

where $w_{i,k}^c$ denotes the $k$-th element of $\mathbf{w}_i^c$. We define the same number of $K$ prototypes for each class. The assignment weights are decided according to the adjacency relationship between the samples and the estimated prototypes, in PALM. Given the probability model in Eq. (1), an embedding $\mathbf{z}_i$ is assigned to a class $c$ with the following normalized probability:

$$p(y_i = c | \mathbf{z}_i, \{\mathbf{w}_i^c, \{\mathbf{p}_k^c, \kappa\}_{k=1}^K\}_{j=1}^C) = \frac{\sum_{k=1}^K w_{i,k}^c Z_D(\kappa) \exp(\kappa \mathbf{p}_k^{c\top}\mathbf{z}_i)}{\sum_{j=1}^C \sum_{k=1}^K w_{i,k}^j Z_D(\kappa) \exp(\kappa \mathbf{p}_k^{j\top}\mathbf{z}_i)}, \tag{10}$$

where $\tau = 1/\kappa$ is analogous to the temperature parameter in our MLE loss. By eliminating $Z_D(\kappa)$, we can obtain Eq. (2) in the main paper as:

$$p(y_i = c | \mathbf{z}_i, \{\mathbf{w}_i^c, \{\mathbf{p}_k^c, \kappa\}_{k=1}^K\}_{j=1}^C) = \frac{\sum_{k=1}^K w_{i,k}^c \exp(\mathbf{p}_k^{c\top} \mathbf{z}_i / \tau)}{\sum_{j=1}^C \sum_{k=1}^K w_{i,k}^j \exp(\mathbf{p}_k^{j\top} \mathbf{z}_i / \tau)}.$$

### A.4 Extending PALM to Unsupervised OOD/Outlier Detection

In this work, we mainly study the OOD detection tasks with the labeled datasets with the class label for each sample. Different from previous methods (Ming et al., 2023; Du et al., 2022a) assigned a single prototype to each class, we automatically maintain and update a mixture of multiple prototypes for each class. Benefiting from PALM's ability to learn multiple prototypes for a set of samples automatically, we study the potential of the proposed techniques to handle the OOD/outlier detection with unlabelled ID training data.

On the unlabelled data, we can define $K$ prototypes $\mathbf{P} = \{\mathbf{p}_k\}_{k=1}^K$ for the entire training dataset without label information and conduct the similar prototype assignment process between all samples and all prototypes. The contrastive loss can be applied to encourage each sample embedding $\mathbf{z}$ to be close to the prototypes based on the assignment weights $\mathbf{w}$ in the form of

$$\ell(\mathbf{z}_i, \mathbf{w}_i) = -\log \sum_{k=1}^K \frac{w_{i,k} \exp(\mathbf{p}_k^\top \mathbf{z}_i / \tau)}{\sum_{k'=1}^K \exp(\mathbf{p}_{k'}^\top \mathbf{z}_i / \tau)}. \tag{11}$$

Without label information, Eq. (11) encourages the sample embeddings to be close to the prototypes based on the weights. Note that the underlying model of assignment weights here is slightly different from the weights in the class-conditional mixture model under the supervised setting. The model also has the potential to consider the relationships between prototypes relying on the distance to produce a latent class-level clustering. The prototype updating and assignment can still be conducted as the standard PALM.

We produce a simple implementation to adapt the proposed method on the unsupervised setting and conduct experiments to validate the potential, as shown in Appendix C.2. Due to the lack of strong supervision information, we utilize data augmentation in implementation, relying on the swapped assignment in (Caron et al., 2020). By letting $\tilde{\mathbf{z}}_i$ denote the embedding of the augmented version of the sample and $\tilde{\mathbf{w}}_i$ denote the corresponding assignment weights obtained for $\tilde{\mathbf{z}}_i$, the loss function based on Eq. (11) can be written as:

$$\mathcal{L}_{\text{unsuper}} = -\frac{1}{N} \sum_{i=1}^N \frac{1}{2} (\ell(\mathbf{z}_i, \tilde{\mathbf{w}}_i) + \ell(\tilde{\mathbf{z}}_i, \mathbf{w}_i)), \tag{12}$$

which is similar to the framework of SwAV for self-supervised learning (Caron et al., 2020). With the experimental results in Appendix C.2, we show a hint of the potential of the proposed techniques for unsupervised OOD detection and will study more detailed design of that in the future works.

## B More Implementation Details

**Software and hardware.** We perform all experiments on an NVIDIA GeForce RTX-3090 GPU using Pytorch.

**Training details for experiments on CIFAR benchmarks.** We utilize ResNet-34 as our backbone model for the CIFAR-100 ID training dataset, and ResNet-18 for CIFAR-10, along with a two-layer MLP projector that projects to a 128-dimensional unit sphere following (Sehwag et al., 2021; Sun et al., 2022; Ming et al., 2023; Tao et al., 2023). We apply stochastic gradient descent with a momentum of 0.9 and weight decay of $10^{-6}$ for 500 epochs. We adopt the same initial learning rate of 0.5 with cosine learning rate scheduling (Loshchilov & Hutter, 2017; Misra & Maaten, 2020), and use the temperature $\tau$ of 0.1 and 0.5 for the MLE loss and prototype contrastive loss, respectively. We set loss weight as 1 to balance the proposed two loss components for simplicity. Following (Caron et al., 2020), we use $\epsilon$ value of 0.05 and 3 *Sinkhorn-Knopp* iterations to obtain our soft

Table 4: OOD detection performance on methods trained on *labeled* CIFAR-10 as ID dataset using backbone network of ResNet-18.

| Methods | OOD Datasets | | | | | | | | | | Average | |
| | SVHN | | Places365 | | LSUN | | iSUN | | Textures | | | |
| | FPR↓ | AUROC↑ | FPR↓ | AUROC↑ | FPR↓ | AUROC↑ | FPR↓ | AUROC↑ | FPR↓ | AUROC↑ | FPR↓ | AUROC↑ |
|---|---|---|---|---|---|---|---|---|---|---|---|---|
| MSP | 59.66 | 91.25 | 62.46 | 88.64 | 51.93 | 92.73 | 54.57 | 92.12 | 66.45 | 88.50 | 59.01 | 90.65 |
| ODIN | 20.93 | 95.55 | 63.04 | 86.57 | 31.92 | 94.82 | 33.17 | 94.65 | 56.40 | 86.21 | 41.09 | 91.56 |
| Vim | 24.95 | 95.36 | 63.04 | 86.57 | 7.26 | 98.53 | 33.17 | 94.65 | 56.40 | 86.21 | 36.96 | 92.26 |
| Energy | 54.41 | 91.22 | 42.77 | 91.02 | 23.45 | 96.14 | 27.52 | 95.59 | 55.23 | 89.37 | 40.68 | 92.67 |
| VOS | 15.69 | 96.37 | 37.95 | 91.78 | 27.64 | 93.82 | 30.42 | 94.87 | 32.68 | 93.68 | 28.88 | 94.10 |
| CSI | 37.38 | 94.69 | 38.31 | 93.04 | 10.63 | 97.93 | **10.36** | **98.01** | 28.85 | 94.87 | 25.11 | 95.71 |
| kNN+ | 2.70 | 99.61 | 23.05 | 94.88 | 7.89 | 98.01 | 24.56 | 96.21 | 10.11 | 97.43 | 13.66 | 97.23 |
| SSD+ | 2.47 | 99.51 | 22.05 | **95.57** | 10.56 | 97.83 | 28.44 | 95.67 | 9.27 | 98.35 | 14.56 | 97.39 |
| NPOS | 4.96 | 97.15 | **17.61** | 91.29 | 3.94 | 97.67 | 13.69 | 95.01 | **7.64** | 94.92 | **9.57** | 95.21 |
| CIDER | 2.89 | 99.72 | 23.88 | 94.09 | 5.75 | 99.01 | 20.21 | 96.64 | 12.33 | 96.85 | 13.01 | 97.26 |
| **PALM** | **0.34** | **99.91** | 28.81 | 94.80 | **1.11** | **99.65** | 34.07 | 95.17 | 10.48 | **98.29** | 14.96 | **97.57** |

online assignment weights. By default, we keep 6 prototypes for each class and select the 5 closest prototypes during the pruning phase. To secure consistent assignment weights across iterations, we set a large momentum $\alpha$ of 0.999 as the default value for prototype updating.

**Details for compared methods.** We follow the standard settings in previous methods (Sehwag et al., 2021; Sun et al., 2022; Ming et al., 2023; Tao et al., 2023) to produce the results of the compared state-of-the-art methods. Details are in the following. For methods that derive OOD detection scores from models trained using standard cross-entropy (*i.e.*, MSP (Hendrycks & Gimpel, 2017), Vim (Wang et al., 2022), ODIN (Liang et al., 2018), Energy (Liu et al., 2020)), we train the model with a initial learning rate of 0.1 and decays by a factor of 10 at epochs 50, 75, and 90 respectively for 200 epochs on CIFAR-100 dataset. We use stochastic gradient descent with momentum 0.9 and weight decay $10^{-4}$. For most recent powerful methods, including VOS (Du et al., 2022b), CSI (Tack et al., 2020), SSD+ (Sehwag et al., 2021), KNN+ (Sun et al., 2022), NPOS (Tao et al., 2023), and CIDER (Ming et al., 2023), we train the model with batch size of 512, using stochastic gradient descent with the same initial learning rate of 0.5 (Khosla et al., 2020; Sehwag et al., 2021; Ming et al., 2023) with cosine learning rate scheduling (Loshchilov & Hutter, 2017; Misra & Maaten, 2020).

**Details for experiments on ImageNet-100.** We use ResNet-50 for ImageNet-100 dataset and project the embeddings to a 128-dimensional unit sphere with a two-layer MLP projector. Following the commonly used setting in OOD detection (Ming et al., 2023), we produce fine-tuning the last residual block and the projection layer on a pre-trained ResNet backbone for 10 epochs, with a learning rate of 0.001.

## C  ADDITIONAL EXPERIMENTAL RESULTS

**OOD Detection Performance on CIFAR-10.** In Table 1, we present the efficacy of PALM against recent competitive OOD detection methods on the challenging CIFAR-100 benchmark. Complementary comparisons on the less challenging CIFAR-10 benchmark are illustrated in Table C. The results indicate that, while recent competitive methods yield comparable performance on this benchmark, PALM continues to achieve results that are on par or even surpass on these OOD datasets. This further demonstrates the efficacy and adaptability of our proposed framework.

**PALM performs superior on Near-OOD detection.** We evaluate the performance of Near-OOD detection following (Tack et al., 2020; Sun et al., 2022), utilizing ID dataset `CIFAR-100` and OOD datasets including `LSUN-FIX`, `ImageNet-FIX`, `ImageNet-RESIZE`, and `CIFAR-10`. While our work is primarily evaluated on the Far-OOD scenario, PALM nonetheless exhibits state-of-the-art performance on these more challenging benchmarks even without explicit considering Near-OOD scenarios. Specifically, PALM yields an improvement of 13.48% in the average FPR and achieves an 81.76 average AUROC score. Importantly, PALM substantially reduces the FPR on ImageNet-R from 56.89 to 27.02, effectively halving the false positives.

**PALM outperforms previous methods using OpenOOD benchmark.** In Table 6, we extend the evaluation of our method using the OpenOOD benchmark Yang et al. (2022); Zhang et al. (2023). Our findings indicate that PALM achieves superior performance on both CIFAR-10 and CIFAR-100

Table 5: *Near-OOD detection* performance on methods trained on *labeled* CIFAR-100 as ID dataset using backbone network of ResNet-34.

| Methods | OOD Datasets | | | | | | | | | |
|---|---|---|---|---|---|---|---|---|---|---|
| | LSUN-F | | ImageNet-F | | ImageNet-R | | CIFAR-10 | | Average | |
| | FPR↓ | AUROC↑ | FPR↓ | AUROC↑ | FPR↓ | AUROC↑ | FPR↓ | AUROC↑ | FPR↓ | AUROC↑ |
| MSP | 88.24 | 69.21 | 86.33 | 70.74 | 86.32 | 72.88 | 88.06 | **76.30** | 87.24 | 72.28 |
| Energy | 87.17 | 72.20 | 78.99 | 76.40 | 80/93 | 80.60 | 86.47 | 70.50 | 84.21 | 74.93 |
| SSD+ | 83.36 | 76.63 | 76.73 | 79.78 | 83.67 | 81.09 | 85.16 | 73.70 | 82.23 | 77.80 |
| KNN+ | 84.96 | 75.37 | 75.52 | 79.95 | 68.49 | 84.91 | **84.12** | 75.91 | 78.27 | 79.04 |
| CIDER | 90.94 | 70.31 | 78.83 | 77.53 | 56.89 | 87.62 | 84.87 | 73.30 | 77.88 | 77.19 |
| **PALM** | **77.15** | **77.24** | **66.19** | **82.51** | **27.02** | **95.03** | 87.25 | 72.28 | **64.40** | **81.76** |

Table 6: OOD detection performance on methods using OpenOOD benchmark.

| Methods | OOD Datasets | | | |
|---|---|---|---|---|
| | CIFAR-10 | | CIFAR-100 | |
| | Near-OOD | Far-OOD | Near-OOD | Far-OOD |
| MSP | 88.03 | 90.73 | 80.27 | 77.76 |
| ODIN | 82.87 | 87.96 | 79.90 | 79.28 |
| Vim | 88.68 | 93.48 | 74.98 | 81.70 |
| Energy | 87.58 | 91.21 | 80.91 | 79.77 |
| OpenGAN | 53.71 | 54.61 | 65.98 | 67.88 |
| VOS | 87.70 | 90.83 | **80.93** | 81.32 |
| CSI | 89.51 | 92.00 | 71.45 | 66.31 |
| kNN | 90.64 | 92.96 | 80.18 | 82.40 |
| NPOS | 89.78 | 94.07 | 78.35 | 82.29 |
| CIDER | 90.71 | 94.71 | 73.10 | 80.49 |
| **PALM** | **92.96** | **98.09** | 76.89 | **92.97** |

datasets compared to previous methods. This further highlight the efficacy and robustness of our approach using the OpenOOD framework.

**More detailed results for the experiments on large-scale OOD detection task.** In Fig. 6 in the main paper, we evaluate the OOD detection performance on large-scale dataset `ImageNet-100` (Tian et al., 2020). Here we provide more detailed performance in Table 7. We fine-tune the last residual block and projection layer of pre-trained ResNet-50 model, while freezing the parameters in the first three residual blocks, following the setting in the previous work (Ming et al., 2023). We compare our approach with recent powerful methods including SSD+ (Sehwag et al., 2021), KNN+ (Sun et al., 2022) and CIDER (Ming et al., 2023). PALM maintains an outstanding performance compared to previous methods, suggesting the effectiveness of our proposed mixture of prototypes framework.

**ID classification accuracy.** We demonstrate the ID classification accuracy in Fig. 7. Fig. 7 shows the ID classification accuaracy of standard cross-entropy (CE), SSD+ (Sehwag et al., 2021) & KNN+ (Sun et al., 2022) (use the same training objective of SupCon (Khosla et al., 2020)), NPOS (Tao et al., 2023), CIDER (Ming et al., 2023), and PALM. Commonly used linear probing (Khosla et al., 2020)

Table 7: OOD detection performance on methods fine-tuning on ImageNet-100 using pre-trained ResNet-50 models.

| Methods | OOD Datasets | | | | | | | | Average | |
|---|---|---|---|---|---|---|---|---|---|---|
| | SUN | | Places | | Textures | | iNatualist | | | |
| | FPR↓ | AUROC↑ | FPR↓ | AUROC↑ | FPR↓ | AUROC↑ | FPR↓ | AUROC↑ | FPR↓ | AUROC↑ |
| SSD+ | **30.34** | 93.06 | **34.38** | 91.52 | 26.49 | 94.84 | 38.19 | 90.96 | **32.35** | 92.60 |
| KNN+ | 41.85 | 92.25 | 44.41 | 90.26 | 26.60 | 94.22 | 38.54 | 94.15 | 37.85 | 92.72 |
| CIDER | 42.26 | 92.84 | 42.81 | 91.39 | 19.31 | 95.44 | 45.49 | 92.83 | 37.47 | 93.12 |
| **PALM** | 42.37 | **93.20** | 41.22 | **91.95** | **17.02** | **96.16** | **32.08** | **95.14** | 33.17 | **94.11** |

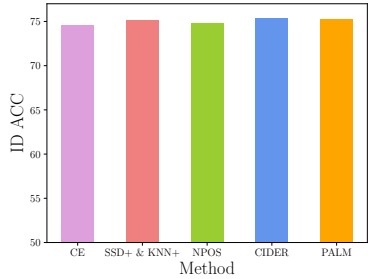

Figure 7: ID classification accuracy of methods trained on CIFAR-100.

Table 8: OOD detection performance comparisons on methods trained on labeled CIFAR-100 as ID dataset using ResNet-34 over 3 independent runs. The experiment results for CIDER are obtained from (Ming et al., 2023).

| Methods | OOD Datasets | | | | | | | | | | | |
| | SVHN | | Places365 | | LSUN | | iSUN | | Textures | | Average | |
| | FPR↓ | AUROC↑ | FPR↓ | AUROC↑ | FPR↓ | AUROC↑ | FPR↓ | AUROC↑ | FPR↓ | AUROC↑ | FPR↓ | AUROC↑ |
|---|---|---|---|---|---|---|---|---|---|---|---|---|
| CIDER | $23.67^{\pm2.28}$ | $95.07^{\pm0.13}$ | $79.37^{\pm1.84}$ | $72.97^{\pm3.90}$ | $22.04^{\pm5.12}$ | $96.01^{\pm1.80}$ | $62.16^{\pm8.48}$ | $83.70^{\pm2.92}$ | $44.96^{\pm6.01}$ | $90.25^{\pm0.97}$ | $46.45^{\pm2.01}$ | $87.60^{\pm1.03}$ |
| PALM | $3.31^{\pm0.34}$ | $99.24^{\pm0.07}$ | $65.51^{\pm1.91}$ | $83.01^{\pm1.66}$ | $10.63^{\pm3.68}$ | $97.83^{\pm0.62}$ | $30.49^{\pm5.83}$ | $94.81^{\pm3.73}$ | $35.45^{\pm2.23}$ | $92.21^{\pm0.31}$ | $31.34^{\pm3.88}$ | $93.02^{\pm1.08}$ |

is used to obtain linear classification accuracy for SupCon, CIDER, and PALM. While producing effective OOD detection, PALM can perform well for ID classification with better or competitive results than other methods, showing the effectiveness of the representation learning.

**Stability of PALM.** To validate the stability of PALM on producing state-of-the-art performance, we train PALM using 3 distinct random seeds and present the average and standard deviation of FPR and AUROC in Table 8. Together with Table 1 in the main paper, it is shown that PALM consistently achieves state-of-the-art performance with a small standard deviation.

## C.1 ADDITIONAL ABLATION STUDIES

**Prototype contrastive loss improves OOD detection capability.** We demonstrate the efficacy of the introduced prototype-level contrastive loss $\mathcal{L}_{\text{proto-contra}}$ in Fig. 8(a) and Table 9. Even without the application of $\mathcal{L}_{\text{proto-contra}}$, our method PALM exhibits state-of-the-art performance. Moreover, by integrating our proposed $\mathcal{L}_{\text{proto-contra}}$ that encourage intra-class compactness and inter-class discrimination at the prototype level, PALM is elevated to a new level of promising performance.

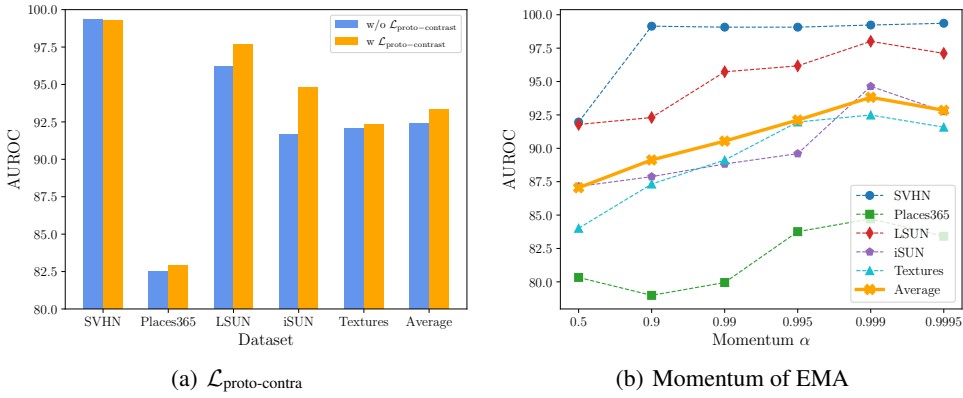

(a) $\mathcal{L}_{\text{proto-contra}}$

(b) Momentum of EMA

Figure 8: Ablation studies on (a) prototype contrastive loss $\mathcal{L}_{\text{proto-contra}}$, (b) the momentum update parameter $\alpha$ of EMA update procedure.

**PALM with different momentum hyperparameters for prototype updating.** In Fig. 8(b), we analyze the influence of maintaining consistent prototype assignment weights. Our observation confirms that utilizing a higher momentum to ensure less changes in the prototypes across iterations, thus achieving more consistent assignment weights, leads to improved performance. The results also show that PALM can perform well consistently with a large enough momentum. Moreover, the momentum parameter $\alpha$ mainly controls the updating strength in the iteration (like a learning rate), which does not influence too much the training behaviours on different data. The experiments show that the influence of $\alpha$ is small. A properly set hyperparameter can work well for many datasets. During the updating of prototypes, the sample-prototype assignment weights are highly relevant to the class/data characteristics. We update with the robust cluster assignment method in Eq. 6 and use the assignment pruning method to alleviate the influence of unexpected points in prototype updating.

<table>
<tr><td colspan="3">Table 9: Ablations on loss weight $\lambda$</td></tr>
<tr><td>Loss Weight $\lambda$</td><td>FPR↓</td><td>AUROC↑</td></tr>
<tr><td>0</td><td>33.77</td><td>92.42</td></tr>
<tr><td>0.1</td><td>33.04</td><td>92.11</td></tr>
<tr><td>0.5</td><td>**27.65**</td><td>93.34</td></tr>
<tr><td>**1 (default)**</td><td>28.02</td><td>**93.82**</td></tr>
</table>

<table>
<tr><td colspan="3">Table 10: Ablations on temperature $\tau$</td></tr>
<tr><td>Temperature $\tau$</td><td>FPR↓</td><td>AUROC↑</td></tr>
<tr><td>0.05</td><td>49.70</td><td>87.51</td></tr>
<tr><td>**0.1 (default)**</td><td>**28.02**</td><td>**93.82**</td></tr>
<tr><td>0.2</td><td>52.88</td><td>84.74</td></tr>
<tr><td>0.3</td><td>76.82</td><td>75.18</td></tr>
</table>

**Ablation on temperature parameter.** In alignment with conventional contrastive learning methods (Chen et al., 2020; Khosla et al., 2020), the temperature parameter $\tau$ plays a crucial role in the training process, serving as a hyperparameter to regulate the compactness of each cluster. It adeptly weights different examples, and selecting an apt temperature is instrumental in aiding the model to learn from hard negatives effectively. Through empirical examination, we empirically found that a temperature of 0.1 yields optimal results.

**Ablation on initialization of prototypes.** In Table 12, we explore the impact of different prototype initialization strategies. Our analysis compares two approaches: initialization with random initialization using a normal distribution, and random initialization with a uniform distribution, the latter being our default setting. The two random initialization variants demonstrate relatively similar performance, showcasing the efficacy of PALM when given proper initialization. This finding further demonstrates the robustness of our approach in updating the prototypes using EMA, showcasing its effectiveness and high-quality prototypes regardless of the initial state of the prototypes.

**Computational efficiency of PALM.** In Table 11 we demonstrate the training speed of our method when training on CIFAR-100 using ResNet-34 with single GeForce RTX3090 GPU, we are comparable as popular contrastive learning method SupCon (Khosla et al., 2020) with negligible overhead. The additional computations of our method are mainly caused by calculating the soft assignment weight using the efficient Sinkhorn-Knopp approximation, which takes only 32 milliseconds each time.

In terms of memory, the additional parameters introduced by the multi-prototype design are only the prototypes as vectors, and the assignment weights for each batch are temporally calculated during training. Under the default setting with 6 prototypes for each of 100 classes (i.e., 600 prototypes in total), it introduces 1.2% of additional parameters compared to the single prototype method. Specifically, based on ResNet-34 with 21.60M, CIDER introduces one prototype per class, increasing the parameters to 21.65M, while our method is 21.91M. The additional computation caused by ours is restricted.

Table 11: Training time for different methods.

| Methods | Training Time (seconds per epoch) |
|---|---|
| CE | 12.34 |
| SupCon | 19.23 |
| CIDER | 23.65 |
| **PALM** | 19.87 |

Table 12: Ablations on the different ways of initializing the prototypes.

| Initialization of Prototypes | OOD Datasets | | | | | | | | | | Average | |
|---|---|---|---|---|---|---|---|---|---|---|---|---|
| | SVHN | | Places365 | | LSUN | | iSUN | | Textures | | | |
| | FPR↓ | AUROC↑ | FPR↓ | AUROC↑ | FPR↓ | AUROC↑ | FPR↓ | AUROC↑ | FPR↓ | AUROC↑ | FPR↓ | AUROC↑ |
| Normal Distribution | 2.70 | 99.39 | 66.04 | 81.33 | 10.66 | 97.77 | 32.97 | 94.26 | 41.08 | 91.08 | 30.69 | 92.77 |
| Uniform Distribution (default) | 3.29 | 99.23 | 64.66 | 84.72 | 9.86 | 98.01 | 28.71 | 94.64 | 33.56 | 92.49 | 28.02 | 93.82 |

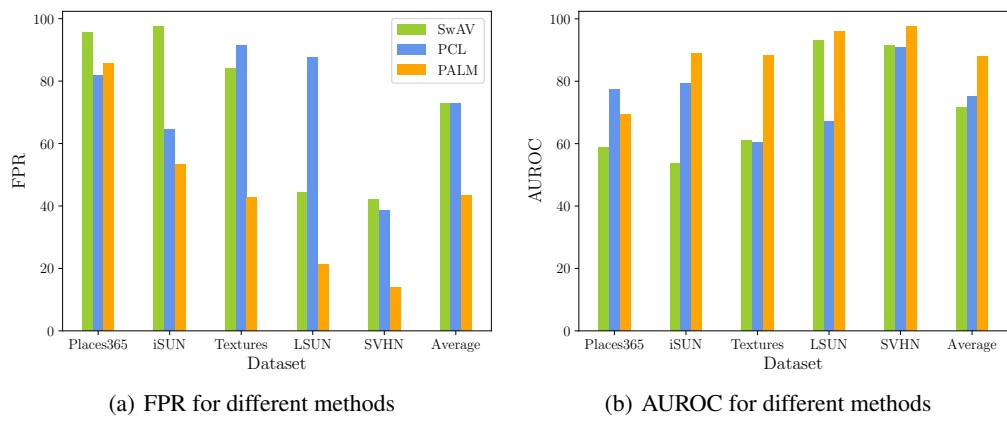

(a) FPR for different methods

(b) AUROC for different methods

Figure 9: OOD detection performance on methods trained on *unlabeled* CIFAR-100 as ID dataset using backbone network of ResNet-34. Smaller FPR values and larger AUROC values indicate superior results.

## C.2 MORE EXPERIMENTS AND DISCUSSIONS ON UNSUPERVISED OOD DETECTION

**Comparisons to previous prototypical contrastive learning approaches on unsupervised OOD/Outlier detection.** In addition to the performance metrics presented in Table 2, we conduct a comparative study between our unsupervised extension and two powerful prototypical contrastive learning frameworks PCL (Li et al., 2021) and SwAV (Caron et al., 2020) on OOD detection performance. All methods are trained on unlabeled CIFAR-100 dataset using ResNet-34, identical to our default experiment settings. For a fair comparison, we do not employ the multi-crop technique suggested in (Caron et al., 2020), we use default data augmentation techniques for all experiments following (Sehwag et al., 2021; Ming et al., 2023) and our default settings.

Compared to PCL (Li et al., 2021), which generates prototype assignments using an offline clustering algorithm that may lead to training instability, our method PALM efficiently assigns prototypes via a stable online procedure. In comparison with SwAV (Caron et al., 2020), we opt to update our prototypes using EMA technique instead of gradient backpropagation, resulting in a more consistent assignment and aiding in the avoidance of sub-optimal solutions. While PCL (Li et al., 2021) and SwAV (Caron et al., 2020) provide similar performance in terms of FPR and AUROC, PALM achieve a **29.35%** improvement on average FPR compared to PCL and an outstanding average AUROC score of **88.07**. The results show the effectiveness of the proposed techniques, rendering promising potential for unsupervised OOD detection.

**EMA technique also benefits unsupervised OOD detection.** In Fig. 4(d), we find that employing EMA technique for prototypes updating considerably benefits the proposed approach PALM for supervised OOD detection. We examine the efficacy of EMA technique for our unsupervised OOD detection extension in Fig. 10. We find that the employed EMA technique remarkably improves our unsupervised extension, achieving state-of-the-art OOD detection performance.

## D MORE DISCUSSIONS ON LIMITATIONS

As discussed in the main paper (in Sec. 5), PALM faces a notable limitation regarding the manual selection of the number of prototypes as a hyperparameter. And we simply assign the same number of prototypes to all classes of samples. Although experiments demonstrated in Fig. 4(c) show that

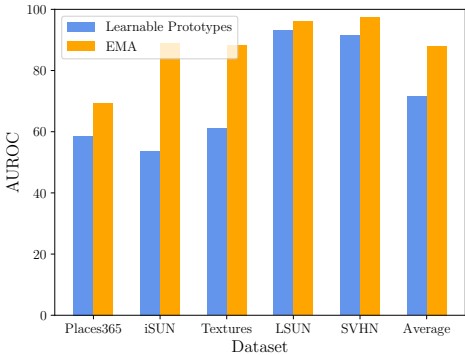

Figure 10: Ablation study on the efficacy of EMA technique for our unsupervised extension.

the PALM can perform well with different numbers of prototypes on the benchmark dataset, it still may limit the applicability of it in more complex scenarios, *e.g.*, the datasets in which different classes require different numbers of prototypes. It may also influence the further extension on the unsupervised setting. In the future, we will consider tackling this limitation by relying on non-parametric Bayesian approaches. Moreover, the potential of the proposed method can be further explored by conducting the learning of the probabilistic mixture model.

