# OpenReview forum: "Learning with Mixture of Prototypes for Out-of-Distribution Detection"
_ICLR.cc/2024/Conference — ICLR 2024 poster_

### Official Review · Reviewer_ddCD · 2023-10-29

**Soundness:** 2 fair
**Presentation:** 3 good
**Contribution:** 2 fair
**Rating:** 6
**Confidence:** 3

**Summary:**

This paper propose a new method that enhances OOD detection by using multiple prototypes per class to capture data diversities, achieving leading results on the CIFAR-100 benchmark.
* PALM introduces a distance-based OOD detection using hyperspherical embedding space and a mixture of prototypes for superior differentiation.
* The system automatically learns prototypes, employing both MLE and contrastive losses to enhance class distinction.
* Empirical studies showcase PALM's effectiveness in both supervised and unsupervised OOD detection scenarios.

**Strengths:**

* The manuscript is articulately presented with clear and coherent language.
* The core idea is presented in a lucid manner, ensuring ease of comprehension for readers.
* The experiments provide compelling evidence for the effectiveness of the proposed approach, e.g., 10-point gain on Places365 (Table 1).

**Weaknesses:**

Major Points:

1. **Hyperparameter Settings:** While it's commendable that ablation studies were performed for the introduced hyperparameters like $K$ (number of prototypes) and $\alpha$ (momentum), how do we choose these hyperparameters for different models or datasets? Is there a potential performance dip with varying settings?

2. **ID Accuracy Impact:** What's the influence of PALM on ID classification accuracy?

3. **Intra-class Heterogeneity:** Fig. 5(c) shows varying prototype effects across datasets. Is this due to different levels of intra-class variability? Do authors have any insights or experiments on why this happens?

4. **Technical Contribution:** The paper's core idea, i.e., prototypical learning with a mixture of prototypes, seems like a straightforward and easy extension of CIDER. I acknowledge the improvements shown in experiments and welcome further discussion on this.

Minor Points:

1. Figure 5 is presented before Figure 4.

2. Figures 4 & 6 are too small to recognize when printed. Consider resizing for clarity.

**Questions:**

Please see the content in Weakness.

---

> ### Author Response · Authors · 2023-11-22
>
> > **Q1. Discussions on hyperparameters for different models or datasets.**
>
> Fig. 5\(c\) and 8(b) show the performance using different hyperparameters $K$ and $\alpha$, which are produced on the benchmark using CIFAR-100 as ID data and using others (such as SVHN, Place365, LSUN, etc) as OOD data. The curves in Fig. 5\(c\) and 8(b) are the performance of the same model trained on the same ID data (e.g., CIFAR-100) on different unseen OOD datasets. They show that the proposed method can perform well and stably given various proper hyperparameters.
>
> Considering the training is only applied on ID data (where OOD data can only be seen in testing), the setting of hyperparameters is mainly for reflecting the ID data characteristics from obtaining high-quality embedding space (with good OOD detection behaviour in testing). In experiments, we observed the method is robust to the hyperparameter setting on different training ID datasets (such as CIFAR-100, CIFAR-10, ImageNet-100). Although tuning the hyperparameters in a proper range may slightly increase or decrease the performance, we use the **same default** hyperparameters to produce the state-of-the-art results in Tables 1-8, demonstrating the robustness of hyperparameter settings on different datasets.
>
> > **Q2. In-distribution classification accuracy.**
>
> Following the settings in KNN, SSD, and CIDER, we demonstrate the ID classification accuracy of the proposed method in Fig. 7. While achieving outstanding OOD detection capabilities, our method also performs well in ID classification. It either surpasses or is competitive with previous methods in this aspect. This highlights the effectiveness of the proposed research, showcasing its proficiency not only in identifying OOD samples but also in accurately classifying ID samples.
>
> > **Q3. Varying behaviours across datasets in Fig. 5\(c\) (i.e., Fig. 4\(c\) in the revised version).**
>
> As discussed in Q1, the curves are for different OOD datasets used in inference, where the model is trained on CIFAR-100 as the ID dataset. It shows that the proposed method can perform well consistently with different settings of the prototypes. Although there may exist an ideal setting for a given training dataset, considering that the main role of prototypes is to regularize/shape the embedding space, a properly set (i.e., not too small or too large) parameter is enough for producing good results.
>
> The behaviours of OOD detection on different testing OOD datasets are slightly different, **mainly due to the difference in data characteristics and the gaps between ID and OOD data**, which is not strongly associated to the intra-class variability in the OOD datasets seen only in testing. For example, while using the place recognition dataset Place365 as OOD data, an OOD input image from Place365 (with a place as the label) can easily contain similar semantic information as the images in CIFAR-100, making OOD detection on it difficult and leading to more unstable results.
>
>
> > **Q4. Highlight the difference to CIDER.**
>
> Thank you for the comments. We highlight the significance of our method and the difference to CIDER.
>
> Among distance-based OOD methods, the main objective is to learn the embedding space (using only ID data) that can discriminate the ID and unseen OOD sample. Different regularizations and supervisions are investigated in various methods based on similar and fair basic frameworks, such as CSI, SSD, CIDER, and our method PALM.
>
> Despite the outstanding performance of CIDER, we identified the limitations of the straightforward (single-prototype) modelling in CIDER. **1)** By analyzing the underlying insights, we first propose to use mixture modelling for better regularizing/shaping the embedding space and thus formulate the mixture vMF models different from CIDER. **2)** Achieving multi-prototype modelling (i.e., moving from a single prototype to multiple prototypes) is not trivial, since, in single prototype modelling, all samples in a class have been assigned to a shared prototype (which is also the simplified assumption). To make multi-prototype modelling achievable, we propose a series of techniques, including prototype updating, sample-prototype assignment considering diversity, and assignment pruning. **3)** Based on that, the novel MLE and contrastive losses based on prototypes use more flexible modelling to shape the embedding space. **4)** Based on the carefully designed modelling and algorithm, the proposed method achieves state-of-the-art performances. We further conducted extensive analyses to analyze the behaviour of the methods. Overall, the relationship between the proposed method and CIDER does not influence the significance of our method.
>
> > **Q5. Minor points for figure presentation.**
>
> Thank you for your suggestion. We have revised the presentation of Fig. 4-7 accordingly. 1) The order of Fig. 4 and 5 is changed. 2) The font size in Fig. 4 and 6 is adjusted to be larger.

---

### Official Review · Reviewer_dG1q · 2023-10-31

**Soundness:** 3 good
**Presentation:** 3 good
**Contribution:** 3 good
**Rating:** 6
**Confidence:** 5

**Summary:**

This paper proposes PALM (PrototypicAl Learning with a Mixture of Gaussian) to address the out-of-distribution detection problem via a distance-based method underlied by a mixture of prototypes. This work is a direct extension of the compactness and dispersion regularized loss (CIDER) by Ming et al. to a probabilistic setting. Although CIDER, and other distance-based OOD detection methods have shown strong results against OOD benchmarks, they indeed tend to make stringent assumptions in their formulation . For instance, CIDER models all samples of each class with only one single prototype, and requires data samples of each class to be compact around a single prototype. These oversimplifying assumptions make the methods quite limiting. This paper addresses some of those limitations to formulate and shape the embedding space.

**Strengths:**

This paper is well-written and mathematically sound. The method PALM that is proposed nicely extends an already strong OOD detection method called CIDER. PALM is extensively analyzed via a thorough ablation study, and extensively evaluated against multiple OOD benchmark datasets and methods. PALM boats strong ID-OOD discrimination in almost all of the experiments by outperforming previous supervised and unsupervised methods by a large margin.

**Weaknesses:**

1. PALM, like its predecessor CIDER, heavily relies on the hyperspherical representation of the learned embeddings to formulate and shape the embedding space. In CIDER, this representation was crucial in achieving strong ID-OOD separability and ID classification. However, since PALM is a mixture of Gaussian, that assumption that the embeddings need to be normalized to unit-norm or need to lie in a hyper spherical space may not necessarily be needed. I wonder if this assumption could be lifted so as to make PALM more general.

2. The second limitation pertains to how the prototypes are updated. While in small data regimes, one can expect the observations to behave similarly in nature, (i.e.; samples of the same class appear somewhat similar throughout the training process), in large data regimes, it's very likely to have samples of the same class behaving differently. In other words, even within the ID distribution, there could be outliers that could affect the ID classification. As a result, maintaining one value of $\alpha$ throughout the training process for the prototypes update seem quite limited to me. One suggestion to the authors is to make $\alpha$ adaptive by integrating its learning as part of the whole training process.

3. It is unclear from the manuscript how the mixture proportions $\omega_{i, k}$ are learned. As the authors may know, GMMs are very sensitive to the initialization of the mixture proportions. A suboptimal initialization may lead to the model getting stuck in a local minimum.

4. It has sounded quite unintuitive to me why the likelihood of a sample would be expressed as a mixture of prototypes given the strong ID separability flavor the authors would want to endow to their ID detection method. This concern is somewhat assuaged by the assignment pruning, but I think more justification is needed as to why one would want to enforce separability and yet mix the prototypes.

5. By having K dedicated prototypes for each class, PALM significantly increases the memory footprint unlike its predecessor CIDER and other distance-based OOD detection methods. This needs to be addressed in the manuscript.

**Questions:**

It'd be great if the authors could address the limitations that I raised above.

---

> ### Author Response · Authors · 2023-11-22
>
> > **Q1. Mixture modeling and normalizing to hyperspherical embedding space**
>
> In our method, we use the mixture of von Mises-Fisher (vMF) distributions (instead of the mixture of Gaussian distributions). The vMF distribution models the points on the hypersphere. Based on this model, we normalize the embeddings to the unit norm. The vMF distribution is commonly used to formulate and explain the embedding space from contrastive learning (Wang & Isola, 2020) ([1] in the following). Based on the contrastive representation learning on the hypersphere (and proper regularizations), the learned embeddings can be discriminative and beneficial for OOD detection, as commonly used in many methods (e.g., SimCLR, SupCpon, CIDER, and PALM) including our PALM. Different from CIDER using single vMF distribution for a class, we use the mixture modelling of vMF to capture the more complex and realistic data.
>
> [1] Wang, T. and Isola, P. Understanding contrastive representation learning through alignment and uniformity on the hypersphere. In International Conference on Machine Learning (pp. 9929-9939). PMLR.
>
> > **Q2. Updating of prototypes.**
>
> Thanks for the comments. Different classes and datasets indeed have different characteristics. A more flexible and dynamic setting of the EMA hyperparameter $\alpha$ can be better but more difficult. However, the momentum parameter $\alpha$ mainly controls the updating strength in the iteration (like a learning rate), which does not influence too much the training behaviours on different data. As shown in Sec. C.1 (Fig. 8(b)), the experiments show that the influence of $\alpha$ is small. A properly set hyperparameter can work well for many cases.
>
> During the updating of prototypes, the sample-prototype assignment weights are highly relevant to the class/data characteristics. We update with the robust cluster assignment method in Eq. (6) and use the assignment pruning method to alleviate the influence of unexpected points in prototype updating.
>
> We revised the paper accordingly.
>
> > **Q3. Estimation of the soft assignment weights $w_{i,k}$.**
>
> To avoid learning a collapsed model, we encourage samples of the same class in a batch to be assigned to diverse prototypes. We optimize $w_{i,k}$'s in $\mathbf{W}^c$ (for each class $c$) by solving the transportation polytope problem in Eq. (8) and (9) (in the revised paper) to maximize the similarity between samples and the assigned prototypes with the regularization. The solution is obtained via Eq. (6). More details of the updating problem are in Sec. 3.3 and Appendix A.2.
> Considering that, although some samples are the top-K neighbours of a prototype, some of them may still be far away like outliers, we further prune the assignment weights for more robust and stable updating, achieving better performance as indicated in Fig. 5(a).
>
> Given the initialized prototypes, the assignment weights (between prototypes and each sample in the batches) in the iterations are calculated relying on the distances and the diversity regularization mentioned above, which avoids the potential trivial and collapsed solution. The prototypes and the subsequent weights are quickly updated to proper positions in the embedding space. Analyses in Appendix C (Tables 7 and 12) show the robustness of initializations (of the prototypes).

---

> > ### Author Response · Authors · 2023-11-22
> > **Continued**
> >
> > > **Q4. Intuitions of mixture modelling and assignment pruning.**
> >
> > The basic idea is to encourage intra-class compactness and inter-class separability for better ID-OOD separability, as other distance-based OOD detection methods. The basic idea is rooted in representation learning, where better ID representation/separability can lead to better handling of unseen/OOD data (Wang & Isola, 2020; Khosla et al., 2020).
> >
> > By using a single prototype to model a class, enforcing all samples (in the class) close to the same prototype in the same way may learn compact intra-class for the majority of samples. However, there may exist outlier samples and diverse sub-classes, which cannot be regularized by a single shared prototype, leading to the mixed embeddings of ID sample and OOD samples, as visualized in Fig. 3(a)(b)\(c\).
> > To address the issue, we introduce a mixture of multiple prototypes for each class and expect any sample can be closely modelled by the associate prototype. Note that we do **not** enforce separability between the intra-class prototypes. Instead, the prototype contrastive loss in Eq. (4) encourages the compactness of the inta-class prototypes and separability of the inter-class prototypes, where the supervision signal is delivered to samples via the prototypes. Multiple prototypes help to capture the diverse characteristics of the samples even in a class, leading to better embedding space (See Fig. 3(d)).
> >
> > As discussed above, during updating prototypes, some samples in selected top-K neighbour samples of a prototype may be too far away from the prototype (but close to another prototype). To approach the mutual neighbour relationship between the assigned samples and prototypes (with limited computations), we use assignment pruning to remove the far-away samples from the top-K neighbour for more reliable prototype updating.
> >
> > > **Q5. Discussions on memory cost.**
> >
> > During the training phase, the only additional memory induced by our proposed method is the storing of multiple prototypes as vectors and the estimated assignment weights temporarily held during training. Compared to CIDER which uses one prototype for each class, the requirements for additional parameters are minimal. We only induce a 1.2% increase in parameters. Specifically, the parameter size for standard ResNet-34 is 21.60M, for CIDER is 21.65M, and for PALM is 21.91M. The paper has been revised accordingly.

---

> > > ### Comment · Reviewer_dG1q · 2023-12-03
> > >
> > > I thank the authors for their rebuttal. Indeed, you are using a mixture of von Mises-Fisher (vMF) distributions in your approach. I missed that. Most of my concerns were addressed. However, the concern I have with the hyper-parameter $\alpha$ remains. Having evaluated CIDER on more complex data, its performance was subpar partly because of how the prototypes are updated. Maintaining one value $\alpha$ throughout the training for large training data regimes doesn't always work. I am not expecting the authors to address this problem in their manuscript, but being aware of this limitation and mentioning it in the paper is important for readers who may be tempted to try out your approach.

---

### Official Review · Reviewer_wPGj · 2023-10-31

**Soundness:** 3 good
**Presentation:** 3 good
**Contribution:** 3 good
**Rating:** 6
**Confidence:** 3

**Summary:**

The paper introduces the PrototypicAl Learning with a Mixture of prototypes (PALM) for Out-of-distribution (OOD) detection in machine learning. Unlike traditional methods that oversimplify data assumptions, PALM models multiple prototypes per class for accurate representations. Experimental results highlight PALM's effectiveness, especially on the CIFAR-100 benchmark.

**Strengths:**

-The paper provides extensive experimental settings, particularly in the ablation studies.
-The proposed method introduces an automatic prototype learning framework that incorporates a mixture of prototypes to represent hyperspherical embeddings, effectively capturing the natural diversities within each class.
-The proposed method achieves a significantly improved performance.

**Weaknesses:**

-I have some concerns about scalability. The introduction of multiple prototypes and their dynamic updating could lead to scalability issues, especially when handling very large datasets or a vast number of classes.
-The effectiveness of PALM is highly dependent on the quality of the prototypes. If the prototypes do not accurately represent the underlying data distribution, the model may face challenges in OOD detection.
-In terms of computational cost, PALM might demand additional computational resources.

**Questions:**

The experimental results from tables 1, 2, and 3 all indicate that PALM does not achieve optimal performance on the texture dataset as it does in other tasks. I am quite curious as to why this phenomenon occurs.

---

> ### Author Response · Authors · 2023-11-22
>
> > **Q1. Discussion on scalability with multiple prototypes and computational cost.**
>
> Although our method involves multiple prototypes per class, it introduces minimal additional overhead on both **computation** and **memory**, not influencing the scalability.
> - In terms of memory, the additional parameters introduced by the multiple-prototype design are only the prototypes as vectors, and the assignment weights for each batch are temporally calculated during training. Under the default setting with 6 prototypes for each of 100 classes (i.e., 600 prototypes in total), it introduces 1.2% of additional parameters compared to the single prototype method. Specifically, based on ResNet-34 with 21.60M, CIDER introduces one prototype per class, increasing the parameters to 21.65M, while our method is 21.91M. The additional computation caused by ours is restricted.
> - The additional computations of our method are mainly caused by calculating the soft assignment weight using the efficient Sinkhorn-Knopp approximation, which takes only 32 milliseconds each time. As discussed in Appendix C.1, the computation time is less than or similar to the previous efficient and powerful methods. With an optimized implementation, our method with multiple prototypes can be faster than CIDER. The computation times are based on the GeForce RTX 3090 GPU.
>
> In the paper, we validated the proposed method with the ImageNet-100 (as ID data used in training), which is a relatively large dataset for OOD, where our method works well and performs better than other methods.
>
> > **Q2. Influence of the quality of prototypes.**
>
> The prototype quality indeed influences the learning. We develop the method with various techniques to ensure the quality of the prototypes, such as the EMA estimation for stable prototype updating and assignment pruning to remove outliers in updating. We observe that, given proper initialization, the method can work effectively. In Appendix C - Table 7, we report the results based on different independent runs with different initializations, having a standard deviation on Average AUROC of only 1.08, implying the robustness of our method in producing high-quality prototypes. To further validate this, we report the results of using different initialization methods for the prototypes in Table 2 in the following. It shows that the proposed method can produce good prototypes and performances in different situations.
>
> **Table 2**: Performance of PALM using different initialization methods for prototypes.
> |  Initialization of Prototypes  |   OOD Datasets  |                 |                 |                 |                 |                 |                 |                 |                 |                 |     Average     |                 |
> |:------------------------------:|:---------------:|:---------------:|:---------------:|:---------------:|:---------------:|:---------------:|:---------------:|:---------------:|:---------------:|:---------------:|:---------------:|:---------------:|
> |                                |       SVHN      |                 |    Places365    |                 |       LSUN      |                 |       iSUN      |                 |     Textures    |                 |                 |                 |
> |                                | FPR| AUROC | FPR | AUROC | FPR | AUROC| FPR | AUROC| FPR | AUROC | FPR | AUROC |
> | Normal Distribution            |       2.70      |      99.39      |      66.04      |      81.33      |      10.66      |      97.77      |      32.97      |      94.26      |      41.08      |      91.08      |      30.69      |      92.77      |
> | Uniform Distribution (default) |       3.29      |      99.23      |      64.66      |      84.72      |       9.86      |      98.01      |      28.71      |      94.64      |      33.56      |      92.49      |      28.02      |      93.82      |
>
> We have added the experiments and discussions in our revised version.
>
> > **Q3. Discussion on performance on Texture dataset in Table 1, 2, and 3 of paper.**
>
> Our method can achieve state-of-the-art results on most of the metrics, especially among the methods with similar frameworks (e.g., distance-based methods), such as CIDER, KNN, and SSD. In Table 1, NPOS achieves the best FPR on Textures as the OOD data, since it directly generates OOD samples to boost training. The Textures dataset, characterized by its diverse textural patterns, may be more amenable to artificial OOD sample generation, especially when compared to the synthesis of natural images, which are more challenging to synthesise. Similarly, in Table 2, our method can perform very well via a simple extension from the labelled setting, while other methods include some specific designs for the unlabelled setting (without semantic labels in the ID training data). In Table 3, we validate the effectiveness of our method with different distance measurements, while the Mahalanobis distance-based version is the best.

---

### Official Review · Reviewer_qjVi · 2023-11-01

**Soundness:** 3 good
**Presentation:** 3 good
**Contribution:** 2 fair
**Rating:** 5
**Confidence:** 3

**Summary:**

This paper presents an outlier detection model, which assumes each class has multiple centroids in feature space, instead of one centroid that many existing models assume. A model called PALM is proposed that minimizes the MLE loss and prototype contrastive loss. The prototype centroid is updated during the model training. Experiment results demonstrate that the proposed model outperforms baselines in OOD and unsupervised OOD tasks.

**Strengths:**

Overall, the novelty of this work is clearly presented, understandable and consistent with the intuition. Experiment comparison is comprehensive.

**Weaknesses:**

There have been some OOD detection benchmark datasets, such as Openood: Benchmarking generalized out-of-distribution detection. Advances in Neural Information Processing Systems 2022, 35, 32598-32611. Most of the datasets used in experiments are based on standard benchmark datasets. How are these datasets, such as CIFAR are used for OOD in this work?

One of the reasons that the proposed model outperforms the compared baselines is it better estimates the class or sample distribution due to the multi-centroid assumption. How about the comparison with generative based models, such as GAN based model?
Reference
Out-of-domain detection based on generative adversarial network. In Proceedings of the 2018 Conference on Empirical Methods in Natural Language Processing (pp. 714-718).

Some technical details are not clearly described. For example, how to have diag(u) and diag(v) in Eqn. (6). Appendix C.1 is not about the detail of this.

**Questions:**

Please refer to the above section.

---

> ### Author Response · Authors · 2023-11-22
>
> > **Q1. Experiments on OpenOOD benchmark. ``Most of the datasets used in experiments are based on standard benchmark datasets. How are these datasets, such as CIFAR are used for OOD in this work?''**
>
>
> Thank you for the suggestion. We evaluate the methods with the OpenOOD benchmark and report the results in the following Table 1. In OpenOOD, while using CIFAR-10 and CIFAR-100 as in-distribution data, other datasets, such as TinyImageNet and MNIST, are used as "Near-OOD" or "Far-OOD". As shown in Table 1, our method is better than or competitive with other methods on most metrics, which validates the effectiveness.
>
> In the paper, the CIFAR datasets are used as in-distribution (ID) data, and other datasets are used as OOD data, which follows the standard experimental protocol widely used in previous works, e.g., CIDER, SSD, Energy, and KNN, as another benchmark. The OOD datasets used in this benchmark are different from OpenOOD. In both the standard and near-OOD experiments in Table 1 and Appendix C - Table 5, our method can achieve the best performance average.
>
> We have added the results on OpenOOD in the paper and updated the discussions.
>
> **Table 1**: Results on OpenOOD benchmarks.
>
> |   Methods   | CIFAR-10 (ID) |           | CIFAR-100 (ID) |           |
> |:-----------:|:---------:|:---------:|:---------:|:---------:|
> |             |  Near-OOD |  Far-OOD  |  Near-OOD |  Far-OOD  |
> | MSP         |   88.03   |   90.73   |   80.27   |   77.76   |
> | ODIN        |   82.87   |   87.96   |   79.90   |   79.28   |
> | Vim         |   88.68   |   93.48   |   74.98   |   81.70   |
> | Energy      |   87.58   |   91.21   |   80.91   |   79.77   |
> | OpenGAN     |   53.71   |   54.61   |   65.98   |   67.88   |
> | VOS         |   87.70   |   90.83   | **80.93** |   81.32   |
> | CSI         |   89.51   |   92.00   |   71.45   |   66.31   |
> | kNN         |   90.64   |   92.96   |   80.18   |   82.40   |
> | NPOS        |   89.78   |   94.07   |   78.35   |   82.29   |
> | CIDER       |   90.71   |   94.71   |   73.10   |   80.49   |
> | PALM (Ours) | **92.96** | **98.09** |   76.89   | **92.97** |
>
> > **Q2. Comparison with generative-based (GAN-based) methods.**
>
> Thank you for the suggestion. We discuss the generative-based (GAN-based) methods [1,2] for OOD detection in the revised paper. Considering the GAN-based method in [1] was originally designed for NLP tasks, we mainly conduct the comparison using the related method OpenGAN [2] on the OpenOOD benchmark, as shown in Table 1. The comparison further provides a broader perspective on the analysis. The results and discussions have been added to the revised version.
>
>
> [1] Ryu, S., Koo, S., Yu, H. and Lee, G.G., 2018. Out-of-domain detection based on generative adversarial network. In Proceedings of the 2018 Conference on Empirical Methods in Natural Language Processing (pp. 714-718).
>
> [2] Kong, S. and Ramanan, D., 2021. Opengan: Open-set recognition via open data generation. In Proceedings of the IEEE/CVF International Conference on Computer Vision (pp. 813-822).
>
> > **Q3. Details of $\text{diag}(\mathbf{u})$ and $\text{diag}(\mathbf{v})$ on Eqn.(6).**
>
> As introduced in Sec. 3.3 and Appendix A.2, $\mathbf{u}$ and $\mathbf{v}$ are two non-negative vectors for obtaining the solution of $\mathbf{W}^c$ constrained by the transportation polytope $\mathcal{W}$ via Eq.(6). As shown in Lemma 2 in (Cuturi, 2013) in the paper, the solution of $\mathbf{W}^c$ is unique and can be obtained via the form of Eq.(6). $\mathbf{u}$ and $\mathbf{v}$ are two non-negative vectors for obtaining the solution, which can be computed with Shinkhorn's fixed point iteration $(\mathbf{u}, \mathbf{v}) \leftarrow (1/(K\mathbf{H}\mathbf{v}),  1/(B\mathbf{H}\mathbf{u}) )$. $\text{diag}(\mathbf{u})$ and $\text{diag}(\mathbf{v})$ are the diagonal matrix of the two vectors $\mathbf{u}$ and $\mathbf{v}$. The detailed proof and algorithms are referred to (Cuturi, 2013). We revised the paper and appendix A.2 with more details accordingly.

---

### Author Response · Authors · 2023-11-22

Dear all reviewers,

Please find the uploaded revised version of our paper. Corresponding to the questions and our responses, we have revised the manuscript and marked the main revisions in blue.

Best regards,

Authors of Submission 1050

---

### Comment · Area_Chair_k2fK · 2023-12-04
**Final Update**

Dear Reviewers,

Please take this chance to carefully read the rebuttal from the authors and make any final changes if necessary.

Please also respond to the authors that you have read their rebuttal, and give feedback whether their rebuttal have addressed your concerns.

Thank you,

AC

---

### Meta-Review · Area_Chair_k2fK · 2023-12-10

**Metareview:**

In this paper, the authors address the out-of-distribution detection problem by proposing a prototypical learning framework with a mixture of prototypes. The proposed method achieves significant improvement over prior art under both supervised and unsupervised settings.

The AC considers that the paper has the following strengths, as also agreed by the reviewers:
+ The paper is overall well-written and easy to follow. The background and problem statement are clear.
+ The motivation and the technical contributions are clear.
+ Literature survey is thorough and the latest status of the field is well-presented.
+ Experiments and results are compelling with comprehensive settings/studies and clear improvements.

The reviewers have raised some concerns and weaknesses in the following aspects:
- Missing benchmark and experiments.
- Missing details.
- Additional memory and compute cost.
- Technical novelty - being a straightforward extension of CIDER.
- Other potential limitations, such as the potential assumption on prototype quality.

Overall the AC finds this paper solid with compelling results. The weaknesses are weaker compared to the overall quality of the paper. In addition, the authors have well-addressed the concerns in their responses, especially the missing benchmark and experiments raised by reviewer qjVi. In this regard, the AC recommends accepting the paper for publication.

**Justification For Why Not Higher Score:**

The findings and novelty of this work is not ground breaking, as reviewers have also pointed out. In addition, the AC considers that the scope of the problem being addressed and its application somewhat limited.

**Justification For Why Not Lower Score:**

While the scores are a bit borderline, the paper is pretty solid and it is clear that the quality is above the threshold. The concerns have been well-addressed and there is no major weakness found in this work.

---

### Decision · Program_Chairs · 2024-01-16

Accept (poster)